# Cloud forcing of surface energy balance from *in-situ* measurements in diverse mountain glacier environments

Jonathan P. Conway[1], Jakob Abermann[2,3], Liss M. Andreassen[4], Mohd Farooq Azam[5], Nicolas J. Cullen[6], Noel Fitzpatrick[7#,] Rianne H. Giesen[8*], Kirsty Langley[3], Shelley MacDonell[9,10], Thomas Mölg[11], Valentina Radic[7], Carleen H. Reijmer[8], Jean-Emmanuel Sicart[12].

1 National Institute of Water and Atmospheric Research, Lauder, New Zealand

2 Department of Geography and Regional Science, University of Graz, Graz, Austria

3 Asiaq – Greenland Survey, 3900 Nuuk, Greenland

4 Section for Glaciers, Ice and Snow, Norwegian Water Resources and Energy Directorate (NVE), Oslo, Norway

5 Department of Civil Engineering, Indian Institute of Technology Indore, India-453552

6 School of Geography, University of Otago, Dunedin, New Zealand

7 Earth, Ocean, and Atmospheric Sciences, University of British Columbia, Vancouver, BC, Canada

8 Institute for Marine and Atmospheric research Utrecht (IMAU), Utrecht University, Utrecht, The Netherlands

9 Centro de Estudios Avanzados en Zonas Áridas (CEAZA), Raúl Bitrán 1305, La Serena, Chile

10 Waterways Centre for Freshwater Management, University of Canterbury and Lincoln University, Christchurch, New Zealand

11 Climate System Research Group, Institute of Geography, University Erlangen-Nürnberg (FAU), Germany

12 Univ. Grenoble Alpes, CNRS, IRD, Grenoble-INP, Institut des Géosciences de l'Environnement (IGE, UMR 5001), F-38000 Grenoble, France

* now at Royal Netherlands Meteorological Institute (KNMI), De Bilt, The Netherlands

# now at Climate Services and Research Applications Division, Met Éireann, Dublin, Ireland

*Correspondence to*: Jonathan P. Conway (jono.conway@niwa.co.nz)

**Abstract.** Clouds are an important component of the climate system, yet our understanding of how they directly and indirectly affect glacier melt in different climates is incomplete. Here we analyse high-quality datasets from 16 mountain glaciers in diverse climates around the globe to better understand how relationships between clouds and near-surface meteorology, radiation, and surface energy balance vary. The seasonal cycle of cloud frequency varies markedly between mountain glacier sites. During the main melt season at each site, an increase in cloud cover is associated with increased vapour pressure and relative humidity but relationships to wind speed are site-specific. At colder sites (average near-surface air temperature in melt season < 0 °C), air temperature generally increases with increasing cloudiness, while for warmer sites (average near-surface air temperature in melt season >> 0 °C) air temperature decreases with increasing cloudiness. At all sites, surface melt is more frequent in cloudy compared to clear-sky conditions. The proportion of melt from temperature-dependent energy fluxes (incoming longwave radiation, turbulent sensible and latent heat) also universally increases in cloudy conditions. However, cloud cover does not affect daily total melt in a universal way, with some sites showing increased melt energy during cloudy

conditions and others decreased melt energy. The complex association of clouds with melt energy is not amenable to simple relationships due to many interacting physical processes (direct radiative forcing, surface albedo, co-variance with temperature, humidity, and wind) but is most closely related to the effect of clouds on net radiation. These results motivate the use of physics-based surface energy balance models for representing glacier-climate relationships in regional- and global-scale

assessments of glacier response to climate change.

## 1 Introduction

Mountain glaciers are sensitive and important components of the climate system. Over the last 50 years, mountain glacier melt has contributed 36-40% of the observed global sea level rise (Hock et al., 2009; Church et al., 2011; Mernild et al., 2014; Zemp et al., 2019; Hugonnet et al., 2021). During the rest of the 21[st] century, a large but uncertain fraction of the remaining mass

stored in mountain glaciers is expected to melt (Radić et al., 2014; Kraaijenbrink et al., 2017; Marzeion et al., 2018; Huss and Hock, 2018; Zekollari et al., 2019). As glaciers are sensitive to change in their surrounding climate, they can be used to infer past changes in climate over decadal (e.g. Mackintosh et al., 2017), centennial (e.g. Oerlemans, 2005; Mölg et al., 2009b) and paleo-climatic timescales (e.g. Putnam et al., 2012).

Our ability to determine how mountain glacier melt responds to changes in climate depends on the ability of models to correctly represent the processes that occur at the atmosphere-glacier interface and link near-surface meteorology and surface melt. The surface energy balance (SEB) is the key process that controls the rate of melt at the glacier surface and can be represented as:

$$Q_M = SWnet + LWnet + Q_S + Q_L + Q_C + Q_{PRC} \qquad\qquad 1$$


where $Q_M$ is the energy available for melt (zero when surface is freezing), $SWnet$ and $LWnet$ are the net fluxes of short and long-wave radiation (including shortwave radiation that penetrates the surface), $Q_S$ and $Q_L$ are the turbulent fluxes of sensible and latent heat, $Q_C$ is the heat flux at the surface from conduction within the glacier  and $Q_{PRC}$ is the heat advected from precipitation. All fluxes are given in (W m$^{-2}$) and those on the righthand side of Equation 1 are defined as positive towards the

surface. When the surface is at the melting point (i.e. surface temperature ($T_s$) = 0 °C), $Q_M$ becomes non-zero and positive, and surface melt ($M$, mm water equivalent) is determined through:

$$M = Q_M * \Delta t / L_f \qquad\qquad 2$$

where $\Delta t$ is the timestep of model output (seconds) and $L_f$ is the latent heat of fusion ($3.34 \times 10^5$ J kg$^{-1}$). In many studies, these relationships between near-surface meteorology and melt are simplified into parameterisations that require less input data such

as temperature index or enhanced temperature index melt models (Huybrechts and Oerlemans, 1990; Hock, 2003; Pellicciotti et al., 2005)

While we know that glaciers are sensitive to changes in local climate, the extent to which cloud cover will amplify or reduce the melting of a glacier in response to future atmospheric warming is uncertain. Clouds alter the incoming shortwave ($SWin$) and longwave ($LWin$) radiation, which are generally the largest sources of energy at the glacier surface (Sicart et al., 2008; Pellicciotti et al., 2011; Van Den Broeke et al., 2011; Cullen and Conway, 2015). Over highly reflective glacier surfaces (e.g.clean snow), a 'radiation paradox' can occur, where net radiation ($Rnet$) increases during cloudy conditions (Ambach,
1974). Clouds can also enhance or dampen the influence of near-surface meteorology, albedo feedbacks and subsurface processes (e.g. refreezing) on SEB and melt (Giesen et al., 2008; Giesen et al., 2014; Conway and Cullen, 2016; Van Tricht et al., 2016; Mandal et al., 2022). As a result, clouds have been associated with both increased and decreased melt rate depending on the climate (Van Den Broeke et al., 2011; Conway and Cullen, 2016; Chen et al., 2021). In the maritime Southern Alps of New Zealand, cloudy conditions have been shown to increase the sensitivity of melt to changes in air temperature (Conway
and Cullen, 2016), due to: (i) more frequent melt in cloudy compared to clear-sky conditions, (ii) increased (positive) $LWnet$ and $Q_L$ in cloudy conditions that enable a similar daily melt rate as clear-sky conditions, and (iii) a change in precipitation phase (from snow to rain) that enhances a positive snowdepth - albedo feedback. The higher sensitivity in cloudy conditions implies that, in the Southern Alps, the response of glacier melt (as well as accumulation) to past and future atmospheric warming will be modulated by atmospheric moisture (in the form of vapour/cloud/precipitation). How these processes interact
in different mountain glacier environments and climate regimes has not been well established.

One challenge has been the lack of direct measurements of cloud amount or type (from e.g. human observer, all-sky camera, or ceilometer) in mountain areas, which has required the derivation of cloud metrics from surface radiation measurements. Studies have employed a variety of methods to derive cloudiness from surface radiation measurements, which limits the ability
to directly compare results from studies in different regions (Giesen et al., 2008; Conway and Cullen, 2016; Sicart et al., 2016; Chen et al., 2021).

The key question of this paper is, therefore: how does cloudiness and its relationships with near-surface meteorology, radiation, and energy balance vary in different mountain glacier environments? The objective is to use a common framework to assess
these relationships at a diverse set of sites where high-quality observations and modelling are available. To guide the analyses, a set of questions was posed:

    i.     How often do different cloud conditions occur at each site?

    ii.    What is the direct effect of clouds on surface radiation at each site?

    iii.   How does near-surface meteorology vary with cloudiness?

iv.   How do the characteristics of melt (e.g. frequency, amount and source of energy) vary in different cloud conditions?

Section 2 sets out the methods used to collate and analyse data sets from 16 glacier automatic weather station (AWS) sites, including the calculation of cloudiness from *LWin*, the definition of melting periods and melt season, and analysis of cloud effects. Section 3 presents results that address the four questions posed above. Section 4 discusses commonalities and differences in cloud – meteorology – SEB – melt relationships, uncertainties and implications for glacier melt modelling.

## 2 Methods

### 2.1 Sites and dataset requirements

Datasets of near-surface meteorology and glacier SEB were collated from a diverse set of sites where high-quality observations and modelling were available. The sites were required to have a published SEB record calculated from AWS data collected over a glacier surface during melt seasons at hourly or smaller timestep. The AWS data needed to include measurements of all four components of the radiation balance, incoming (*SWin*) and outgoing shortwave (*SWout*), incoming (*LWin*) and outgoing longwave (*LWout*), all in W m$^{-2}$. In addition, turbulent fluxes were to be calculated using bulk aerodynamic methods avoiding potentially inaccurate assumptions (e.g. surface temperature fixed at 0 °C regardless of SEB). Note that published values of surface melt and SEB fluxes are used in these analyses rather than being recalculated from near-surface meteorology and radiation. Thus, differences in the methods used to calculate SEB may introduce some uncertainty (mainly in the calculation of sub-surface fluxes), but the values are congruent with previous studies, and no additional validation is needed. A call for datasets was made on *Cryolist* in January 2020, and data from over 30 sites was offered. After assessing each dataset against the criteria above, 16 sites were selected for analysis (Figure 1 and Table 1). These sites covered many of the mountain glacier regions including continental North America, the European Alps, Norway, Greenland, the Himalaya, tropical glaciers in Africa and the Andes, the arid region of central Chile and the Southern Alps of New Zealand. It is worth noting that no suitable datasets were made available from some large regions of mountain glaciers including Alaska, Patagonia and Asia outside of the Himalaya.

As most AWS sites are in ablation areas, they follow a broad pattern of decreasing altitude with distance from the equator (Figure 2). Note that two locations have observations in both the ablation and accumulation area - Conrad Glacier (CABL, CACC) and Mera Summit (MERA) / Naulek (NAUL, an ablation area of Mera Glacier). Records from the same site in different years were also joined into continuous records (CABL and NAUL). Records from CABL, CACC and NORD cover only summer periods and CHHO has three two-month periods throughout the year, otherwise the records span all months of the year and range from 46 to 3231 days in length (See Table 1 for site name abbreviations). Figures A1 and A2 show monthly average meteorology and SEB fluxes for each site used in the analysis. A few broad groupings of sites (listed in Table 1) can be identified through seasonal trends in near-surface air-temperature ($T_a$; °C) or relative humidity (*RH*) in Figure A1: mid- and high- latitude maritime and continental sites with strong seasonal cycles of $T_a$ but small variations in *RH*; Himalayan sites with

strong cycles of $T_a$, and distinct wet and dry seasons; tropical sites with small variations in $T_a$ and distinct wet and dry seasons; and a mid-latitude arid site (GUAN) with low *RH*.


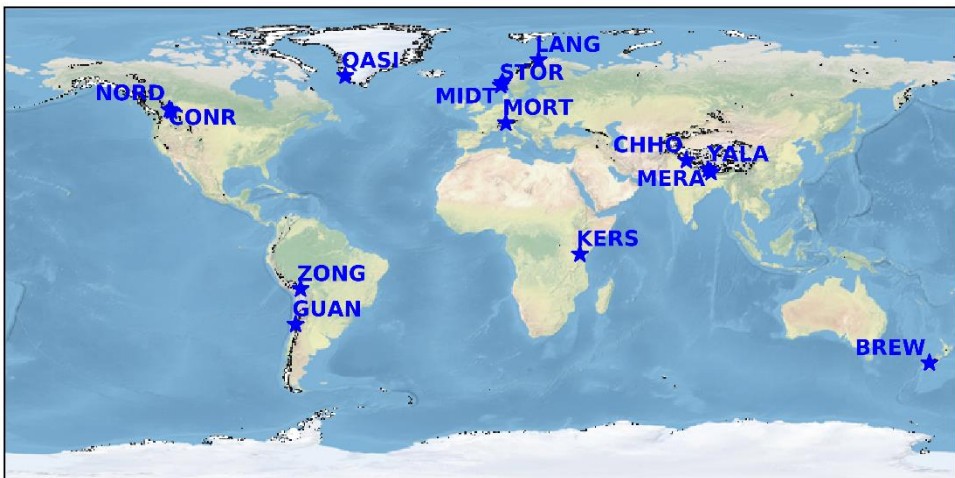

**Figure 1: Map showing location of study sites with short names (See Table 1 for full names) along with glacier areas from the Randolph Glacier inventory (black outlines; RGI Consortium, 2017). Note the two Conrad Glacier sites (CABL, CACC) are shown**
**as CONR and the two Mera Glacier sites (MERA, NAUL) as MERA. The background map is Natural Earth shaded relief.**

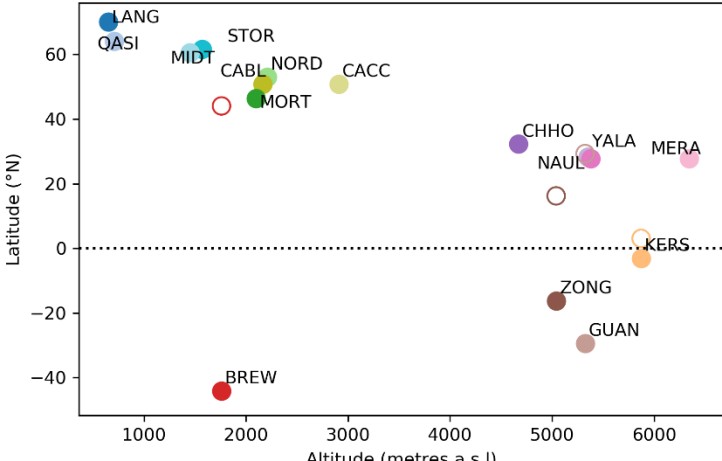

**Figure 2: Altitude and latitude of study sites. Open circles show the position of southern hemisphere sites against northern hemisphere sites for comparison.**


**Table 1: Details of study sites listed by latitude**

| Name | Short name | Latitude (°N) | Longitude (°E) | Altitude (m) | Regional climate grouping | Record length (days) | Years of record | Reference |
|------|-----------|--------------|---------------|-------------|--------------------------|---------------------|----------------|-----------|
| Langfjordjøkelen | LANG | 70.133 | 21.75 | 650 | High-lat. maritime | 1070 | 2007-10 | Giesen et al. (2014) |
| Qasigiannguit | QASI | 64.162 | -51.359 | 710 | Mid-lat. maritime | 703 | 2014-16 | Abermann et al. (2019) |
| Storbreen | STOR | 61.583 | 8.166 | 1570 | Mid-lat. maritime | 1827 | 2001-06 | Andreassen et al. (2008); Giesen et al. (2009) |
| Midtdalsbreen | MIDT | 60.567 | 7.467 | 1450 | Mid-lat. maritime | 2137 | 2000-06 | Giesen et al. (2008); Giesen et al. (2009) |
| Nordic | NORD | 53.051 | -120.444 | 2208 | Mid-lat. continental | 46 | 2014 | Fitzpatrick et al. (2017) |
| Conrad (ablation) | CABL | 50.823 | -116.920 | 2164 | Mid-lat. continental | 119 | 2015-16 | Fitzpatrick et al. (2019) |
| Conrad (accum) | CACC | 50.782 | -116.912 | 2909 | Mid-lat. continental | 68 | 2016 | Fitzpatrick et al. (2019) |
| Morteratsch | MORT | 46.422 | 9.9318 | 2100 | Mid-lat. continental | 3231 | 1998-2007 | Oerlemans et al. (2009) |
| Chhota Shigri | CHHO | 32.28 | 77.58 | 4670 | Himalaya Monsoon-arid transition | 177 | 2012-13 | Azam et al. (2014) |
| Yala | YALA | 28.235 | 85.618 | 5350 | Himalaya Monsoonal | 811 | 2014-18 | Litt et al. (2019) |
| Mera Summit | MERA | 27.706 | 86.874 | 6342 | Himalaya Monsoonal | 867 | 2013-16 | Litt et al. (2019) |
| Naulek (Mera) | NAUL | 27.718 | 86.897 | 5380 | Himalaya Monsoonal | 1387 | 2013-17 | Litt et al. (2019) |
| Kersten | KERS | -3.078 | 37.354 | 5873 | Tropical | 1078 | 2005-08 | Mölg et al. (2009b) |
| Zongo | ZONG | -16.25 | -68.167 | 5040 | Tropical | 362 | 1999-2000 | Sicart et al. (2005) |
| Guanaco | GUAN | -29.34 | -70.01 | 5324 | Mid-lat. arid | 910 | 2008-11 | MacDonell et al. (2013) |
| Brewster | BREW | -44.08 | 169.43 | 1760 | Mid-lat. maritime | 676 | 2010-12 | Conway and Cullen (2016); Cullen et al. (2016) |

## 2.2 Data processing

Data from each site were taken through several processing steps as outlined in Figure 3. After basic quality control and homogenisation (described below), a timeseries of cloudiness was generated for each site (Section 2.3), melting periods and the main melt season were defined (Section 2.4), after which cloud effects on melt were analysed (Section 2.5).

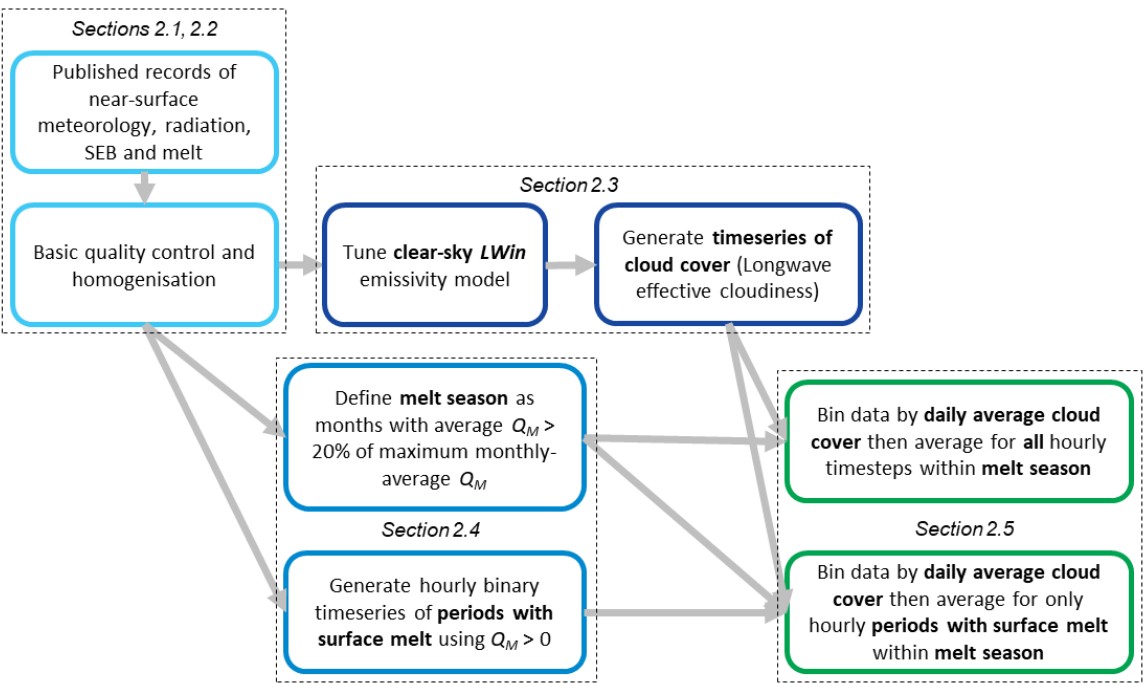

**Figure 3: Steps used to process and analyse data, annotated with relevant sections of the methods.**

Basic quality control and homogenisation involved the following steps:
- Sub-hourly data resampled to hourly time steps
- Times converted to local solar time using longitude rounded to nearest full hour offset from UTC.
- Data cut to full days only (no days with partial missing data)
- Naming, units and sign conventions of variables standardised
- Periods with missing radiation data ($SWin$, $SWout$, $LWin$, $LWout$) removed
- Periods with missing $T_a$ and $RH$ data removed.
- Negative values of $SWin$ and $SWout$ set to 0
- Values of $LWout > 315.6$ W m$^{-2}$ reset to 315.6 W m$^{-2}$
- Net radiation ($Rnet$) calculated from corrected values of ($SWin$, $SWout$, $LWin$, $LWout$)

- Near-surface vapour pressure ($e_a$; hPa) calculated from $T_a$ and $RH$ using Buck (1981)

- Surface temperature ($T_s$; °C), if not provided, calculated from $LWout$ using the Stefan-Boltzmann law and a surface emissivity of 1

- Daily average albedo calculated as ratio of daily sums of $SWin$ and $SWout$

- If $Q_M$ or surface melt calculated from SEB model is not provided, then $Q_M$ is calculated as positive values of SEB when $T_s > -0.1$ °C. The slightly relaxed constraint on $T_s$ allows for some uncertainty in measured $T_s$.

Monthly statistics (averages, frequencies by bin etc.) were only calculated when at least 10 days of data from a given month were available.

**2.3 Defining clear-sky and cloudy periods using incoming longwave radiation**

For each site, timeseries of cloudiness were derived from measured $LWin$, $e_a$ and near-surface air temperature ($T_{a,K}$; K) following Konzelmann et al. (1994) and Conway et al. (2015). First, the effective sky emissivity ($\varepsilon_{eff}$) was calculated using:

$$\varepsilon_{eff} = LWin/\sigma T_{a,K}^{4}$$   3


where σ is the Stefan–Boltzmann constant ($5.67 \times 10^8$). While $LWin$ is influenced by emission from surrounding terrain, the sky-view factor at all sites is close to 1 and horizons at all sites are below the limit of the sensor field of view, so no corrections were needed here.

Timeseries of theoretical clear-sky emissivity ($\varepsilon_{cs}$) at each site were defined using the Brutsaert (1975) curve as modified by Konzelmann et al. (1994) with the exponent set to 1/7 after Dürr et al. (2006):

$$\varepsilon_{cs} = \varepsilon_{ad} + b\left(100 \times e_a/T_{a,K}\right)^{(1/7)}$$   4

where $\varepsilon_{ad}$ is an elevation-dependent dry air emissivity term (varying between 0.18 and 0.23) defined here using $\varepsilon_{ad}$ values determined from radiative transfer modelling in Durr et al. (2006) for the European Alps that are regressed against elevation ($z$; m above sea level):

$$\varepsilon_{ad} = 0.2351 - z \times 9.636 \times 10^{-6}$$   5


For each site, Equation 4 was fitted to the lowest 10% of $LWin$ in each of 30 $e_a/T_{a,K}$ bins (Figure A3) by finding the value of $b$ (in 0.001 steps) that gave the smallest root mean square error (RMSE). This step used only hours with valid $LWin$, $e_a$ and $T_{a,K}$ values and $RH < 80\%$. Optimised values of $b$ and RMSE are given in Table A1.

Timeseries of longwave equivalent cloudiness ($N_\varepsilon$) were then derived by fitting hourly measured $\varepsilon_{eff}$ between theoretical clear-sky ($\varepsilon_{cs}$) and overcast ($\varepsilon_{ov} = 1$) emissivity values, limiting $N_\varepsilon$ to a range 0 to 1 (Conway et al., 2015):

$$N_\varepsilon = (\varepsilon_{eff} - \varepsilon_{cs})/(\varepsilon_{ov} - \varepsilon_{cs}) ; \qquad\qquad\qquad\qquad 6$$
$$N_\varepsilon[N_\varepsilon > 1] = 1; N_\varepsilon[N_\varepsilon < 0] = 0$$

Following Giesen et al. (2008), clear-sky conditions are defined as $N_\varepsilon <= 0.2$, partial-cloud as $0.2 < N_\varepsilon < 0.8$ and overcast as $N_\varepsilon >= 0.8$. Daily average, rather than hourly average, $N_\varepsilon$ was used to define cloudiness to reduce noise, limit the influence of diurnal cycles in variables and focus on synoptic scale (daily) variability in cloud – SEB relationships. Note that moderate values of daily average cloudiness can indicate either patchy cloud cover and/or a mix of overcast and clear-sky conditions during a day. Cloudiness can be derived from *SWin* (e.g. Greuell et al., 1997; Sicart et al., 2006; Mölg et al., 2009a; Kuipers Munneke et al., 2011) but was considered a less appropriate metric here as its calculation relies onsetting a typical cloud extinction coefficient that differs between sites (Pellicciotti et al., 2011). In addition, cloudiness cannot be derived from *SWin* during the night and terrain shading of *SWin* introduces further uncertainty, especially in winter..

## 2.4 Definition of melt season and periods with surface melt

For each site, a melt season was defined as the months in which monthly-average $Q_M$ at the site was greater than 20% of the maximum monthly-average $Q_M$ for the same site (Figure A2; A4). This proved a simple method to retain months with substantial melt but exclude winter months where melt is infrequent. The sensitivity of this choice was assessed by replicating key results using only months with monthly-average $Q_M$ greater than 80% of the maximum monthly-average $Q_M$ for that site. Rather than only selecting individual melt events for analysis, averages over all timesteps in the melt season were used to better understand the relationships between cloudiness, surface radiation and near-surface meteorology, without skewing the data towards melt episodes that may have atypical meteorology. To identify the times surface melt occurred and to quantify the contributions of SEB components to $Q_M$, periods with surface melt were defined as hourly timesteps with $Q_M > 0$.

## 2.5 Analysis of cloud effects

The relationship between cloudiness, meteorology, SEB and melt is assessed by binning the timeseries of different variables by daily average cloudiness. Five evenly sized bins were used with bin centres at $N_\varepsilon = 0.1, 0.3, 0.5, 0.7$ and $0.9$, with the top and bottom bins corresponding to clear-sky and overcast conditions, respectively. Data within each bin were then averaged across all days within the main melt season to demonstrate the average relationships between cloudiness and different variables.

In sections 3.2, 3.3 and 3.4, we use the term cloud effects to describe the change in a variable during cloudy conditions with respect to clear-sky conditions. In studies of net radiation, the cloud effect (CE) is defined as the difference between average and clear-sky conditions (e.g. Ambach, 1973; van den Broeke et al., 2008). Here we extend the concept to $Q_M$ in order to describe the average change in melt related to clouds, even though clouds are not the only meteorological forcing responsible for changes in $Q_M$. We calculate CE for all net radiation components (*SWnet, LWnet, Rnet*) and $Q_M$. Here, we calculate CE by

subtracting the average value in the clear-sky bin ($N_\varepsilon <= 0.2$) from the average value equally weighted across all cloudiness bins. Equally weighting each cloudiness bin ensures that differences in the frequency of different cloud conditions do not skew the data between sites.

## 3 Results

### 3.1 Cloud metrics

#### 3.1.1 Effective sky emissivity and fitted clear-sky curve

The derivation of clear-sky emissivity from *LWin* highlighted substantial variations in the relationship between near-surface meteorology and *LWin* between the sites. On an hourly basis, most sites show a preference for either clear-sky or overcast conditions, as shown by the darker colours around the clear-sky and overcast emissivity (Figure 4). Sites in the Himalaya (CHHO, YALA, NAUL, MERA) showed a distinct seasonality with predominately warm/wet/overcast or cold/dry/clear-sky

conditions. Tropical and arid glacier sites (KERS, GUAN) show a much lower $\varepsilon_{cs}$ for the same surface vapour pressure, in part due to the high elevation (therefore low $\varepsilon_{ad}$), but also due to the low value of *b* (Equation 4; Table A1), which indicates a thinner atmospheric water vapour profile above the surface compared to Himalayan sites at similar altitudes. Mid-latitude sites with records covering the full annual cycle in Europe (LANG, MIDT, MORT, STOR) and New Zealand (BREW) show a similar preference for cold/dry/clear-sky or warm/wet/overcast conditions, while QASI shows a greater frequency of cloud at

lower temperature/vapour pressure. Sites in the Western Cordillera of Canada (NORD, CABL, CACC) and Europe (MIDT, MORT, STOR) show more frequent partial cloud than many other sites. Note that the short summertime records from Canada (NORD, CABL, CACC) do not capture the full spectrum of conditions at these sites.

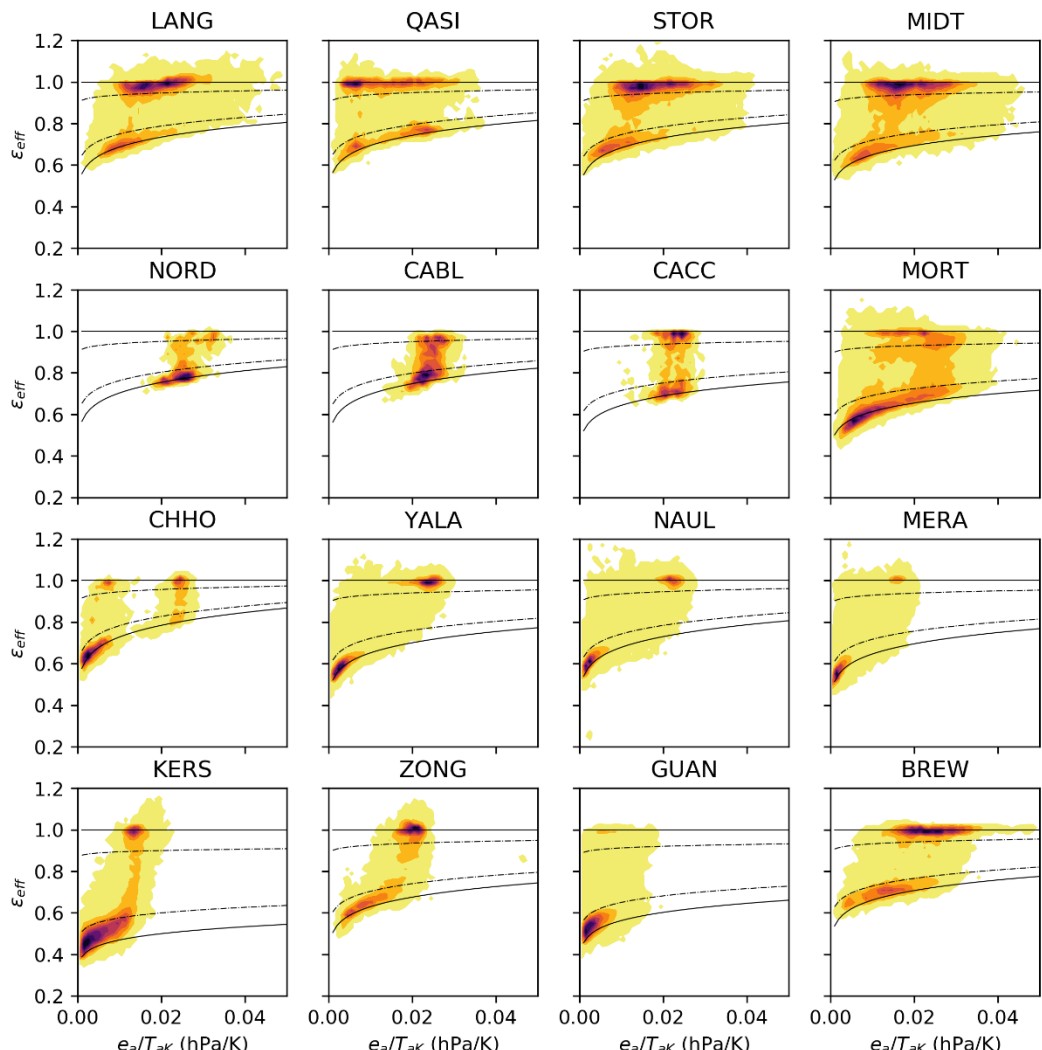

Figure 4: Frequency of observed $\varepsilon_{eff}$ (filled contours) versus $e_a/T_{a,K}$ for sites arranged by latitude. Also shown are calculated $\varepsilon_{cs}$ (lower solid line), $\varepsilon_{ov}$ (upper solid line) and $\varepsilon_{eff}$ at clear-sky and overcast limits of $N_\varepsilon = 0.2$ and $N_\varepsilon = 0.8$, respectively (lower and upper dashed lines, respectively). Contours of relative frequency created from 2D histogram with common x and y bins across all sites with colours in 10 steps between 1 (yellow) and the maximum number of hours in any x, y bin for each site (dark brown/black).

### 3.1.2. Monthly cloud frequency

The frequency of clear-sky, partial-cloud and overcast conditions also shows distinct regional and seasonal variations (Figure 5 for daily average, Figure A4 for hourly periods). Mid-latitude glaciers in maritime locations show very limited seasonality (BREW, STOR, MIDT) and a high percentage of overcast conditions, except for LANG that displays more frequent overcast conditions during the melt season and QASI that shows a tendency towards more frequent clear-sky conditions during its melt

season. Mid-latitude sites in continental locations (NORD, CABL, CACC, MORT) show less frequent overcast and more frequent partial-cloud conditions than the mid-latitude maritime sites, with MORT showing more frequent partial-cloud conditions during the melt season and more frequent clear-sky conditions in the winter. Most Himalayan sites (YALA, MERA, NAUL) show much stronger seasonality, with more frequent overcast conditions during the melt season. The exception is CHHO, which shows weaker monsoon influence (fewer overcast conditions) being on the transition zone between monsoon and arid regions (Azam et al., 2021), though the fraction of partial-cloud conditions still increases in July and August. While ZONG experiences melt most of the year, melt rates are higher during the cloudier months from September through April corresponding with marked seasonal changes in cloud and SEB caused by the tropical climate (Figure A2). KERS experiences less cloud from June through October, with low melt rates year-round. GUAN experiences the least cloud, with predominately clear-sky conditions and only sporadic melt during austral summer.

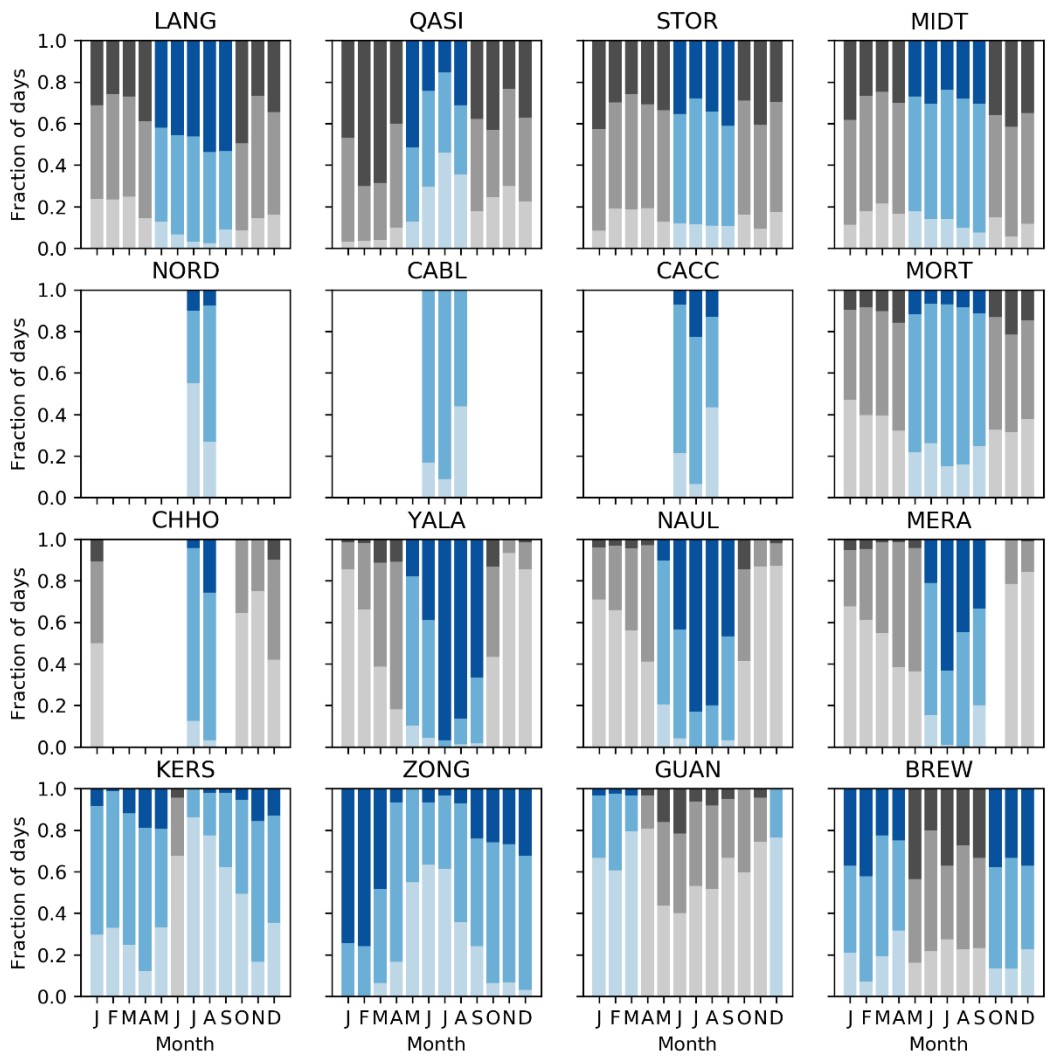

**Figure 5: Monthly fraction of clear-sky (light shading), partial-cloud (mid shading) and overcast conditions (dark shading) defined using daily average cloudiness ($N_\varepsilon$). Months defined as within the 'melt season' are shaded blue.**

## 3.2 Cloud effects on melt-season surface radiation

An estimate of the direct effect of clouds on the SEB is gained by examining the variation of incoming radiation (*SWin* and *LWin*) with cloudiness (Figure 6). At most sites the average direct effect of clouds on incoming radiation is negative, steadily decreasing with increasing cloud cover to between -60 and -170 W m$^{-2}$ (Figure 6f). The exceptions are low-latitude and high-altitude sites KERS, MERA, and ZONG, where comparatively small decreases in *SWin* with cloudiness (Figure 6d) are compensated by large increases in *LWin* (Figure 6e). The large variation in *SWin* and *LWin* cloud effects between sites suggests

that different cloud types and cloud properties play a role in determining radiative forcing and this should be investigated in future work. We note that changes in the profile of water vapour and air temperature (estimated by $e_a$ and $T_a$) also influence *LWin* (and to a much lesser extent *SWin*). Hence, the direct cloud effects shown here represent the combined effects of direct radiative forcing and changes to atmospheric profiles of water vapour and temperature, in contrast to analyses of cloud radiative forcing that consider the changes in incoming radiation with respect to calculated clear-sky values (e.g. Sicart et al., 2016).

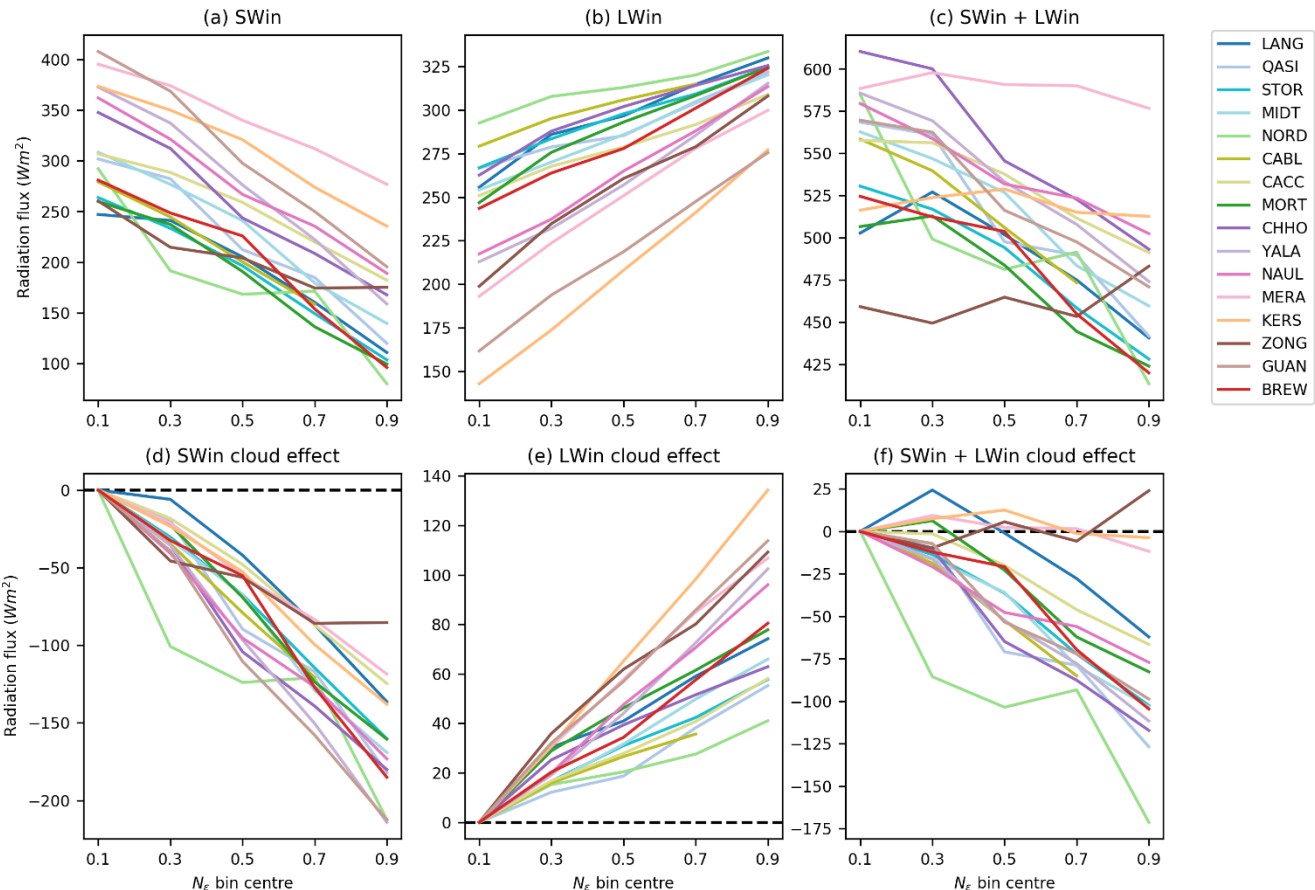

**Figure 6: (a)-(c) Average melt-season incoming radiation fluxes (*SWin, LWin*) for different daily average cloud conditions ($N_\varepsilon$), (d)-(f) as for (a)-(c) expressed as change from clear-sky conditions ($N_\varepsilon <= 0.2$). Note y-axis range differs between panels.**

By analysing the change in net radiation fluxes (*SWnet, LWnet* and *Rnet*) the effect of albedo and surface temperature is included with the direct effect of clouds on incoming radiation (Figure 7). A clear increase in *Rnet* during cloudy periods (positive *Rnet* cloud effect), aka 'radiation paradox', is observed at some sites: ZONG, MERA, LANG (Figure 7f), due to small negative *SWnet* effect and strong positive *LWnet* effect (Figure 7d,e). GUAN and KERS have a similarly strong positive *LWnet* effect at higher values of $N_\varepsilon$, but much more negative *SWnet* effects cancel these out. For most sites, the *Rnet* cloud

effect is small and negative (0 to -20 W m$^{-2}$). Many of these sites show a decrease in *Rnet* only at higher values of $N_\varepsilon$, while 3 sites (MIDT, MORT, CHHO) show the highest *Rnet* in partial-cloud conditions, emphasising that the relationship between *Rnet* and cloudiness is not always linear. NORD, CABL, QASI, and CHHO all show a strong negative *Rnet* cloud effect, driven by strong negative *SWnet* effect and weak *LWnet* cloud effect. For the two sites with measurements from both the accumulation and the ablation areas, accumulation sites exhibit more positive and/or less negative *Rnet* cloud effect compared with their ablation area counterparts, driven by the change in *SWnet* cloud effect (surface albedo) rather than a large change in *LWnet* cloud effect.

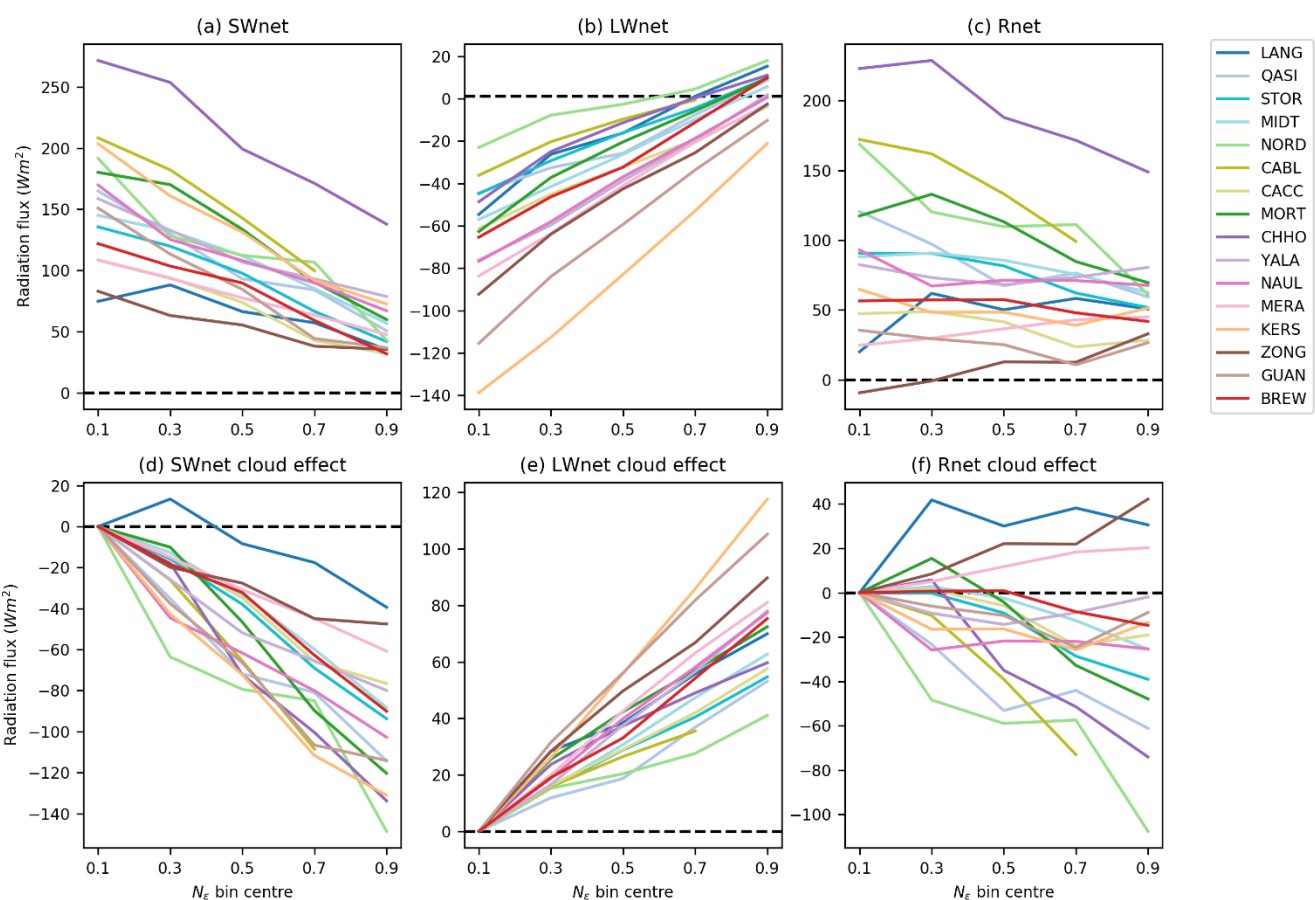

**Figure 7: (a)-(c) Average melt-season net radiation fluxes (*SWnet, LWnet, Rnet*) for different daily average cloud conditions ($N_\varepsilon$), (d)-(f) as for (a)-(c) expressed as change from clear-sky conditions ($N_\varepsilon <= 0.2$). Note y-axis range differs between panels.**

### 3.3 Variation of near-surface meteorology with cloudiness

Alongside radiative changes, differences in near-surface meteorology are also an important driver of SEB and melt variations
with cloudiness, particularly $Q_S$, $Q_L$ and $LWin$. Air temperature shows a divergent relationship to cloudiness; at sites with
average melt-season $T_a \gg 0$ °C, increasing cloudiness is associated with lower temperatures, while at sites with average melt-
season $T_a < 0$ °C (KERS, MERA, NAUL, YALA), cloudiness is generally associated with higher temperatures (Figure 8a).
Average $T_a$ varies little with cloud cover at ZONG and CHHO. At most sites, wind speed decreases with increasing cloudiness
(Figure 8b). The exceptions are BREW and STOR, which show moderate increases (< 1 m s$^{-1}$), LANG and MIDT, which show
larger increases (1.6 and 2.9 m s$^{-1}$, respectively), QASI, which shows no large change cloudiness and CACC, which shows
peak wind speed at moderate cloudiness . Sites where wind speed increases with cloudiness (particularly MIDT and LANG)
have a wind climate that is mainly influenced by the large-scale circulation, while other sites may have a more local wind
climate where local or meso-scale katabatic or convective circulations prevail (e.g. Mölg et al., 2020; Conway et al., 2021).
Stronger radiative cooling during clear-sky periods may promote higher katabatic wind speeds in clear-sky conditions, though
the relationship is not simple; at ZONG, strong winds during clear-sky conditions are related to large-scale forcing during the
dry season (Litt et al., 2014). As expected, $e_a$ and $RH$ increase with cloudiness, however some sites with $e_a$ around the saturation
vapour pressure of melting surface show a weak relationship to cloudiness (e.g. QASI, CACC). The wide variation of $RH$ in
clear-sky conditions (~30 to ~70%) implies that care should be taken when using $RH$ to model cloud cover using empirical
parameterisations developed for particular study areas, or even at different altitudes (e.g. NAUL vs MERA).

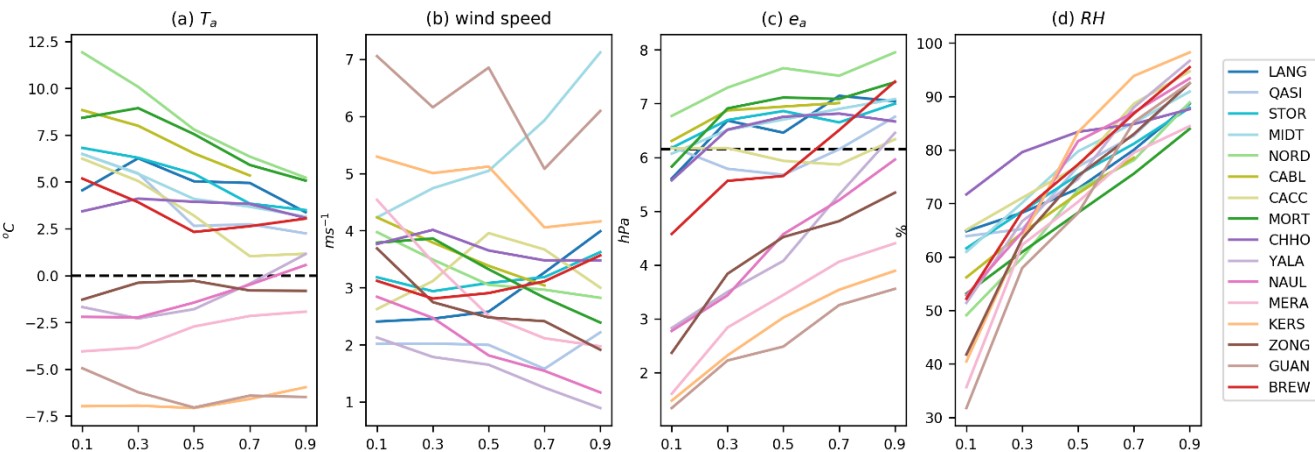

**Figure 8: Average melt-season near-surface meteorology for different daily average cloud conditions ($N_\varepsilon$). Dashed lines indicate melting point temperature in (a) and saturation vapour pressure in (c).**

## 3.4 Variation of melt frequency, melt amount and SEB with cloudiness

The percentage of hours with surface melt increases with cloudiness at all study sites (Figure 9), with the exception of GUAN, which experiences very infrequent melt in all conditions. Colder sites across the Himalaya and tropical regions (except KERS) show the largest increases with respect to clear-sky conditions (up to 5 times more frequent), while BREW, MORT and LANG all show moderate increases up to 1.5 times more frequent in overcast conditions. Other European and North American sites show comparatively high melt frequency across all cloud conditions, indicative of the warm conditions where $e_a$ exceeds that of a melting ice/snow surface. Even in these conditions, periods with surface melt still become more common with increasing cloudiness, with 100% of overcast periods at NORD experiencing melt (Figure 9a). While analysis of diurnal patterns of melt is beyond the scope of this paper, the higher percentage of hours with melt during overcast conditions indicates that night time melt is more frequent during overcast periods. . MERA shows the largest increase in melt frequency with cloudiness, with melt 5 times more frequent in overcast (26% of overcast conditions) compared to clear-sky conditions (5%). A consistent increase with cloudiness is observed at MERA but caution is warranted given the small number of hours with melt in clear-sky conditions (20 hours).

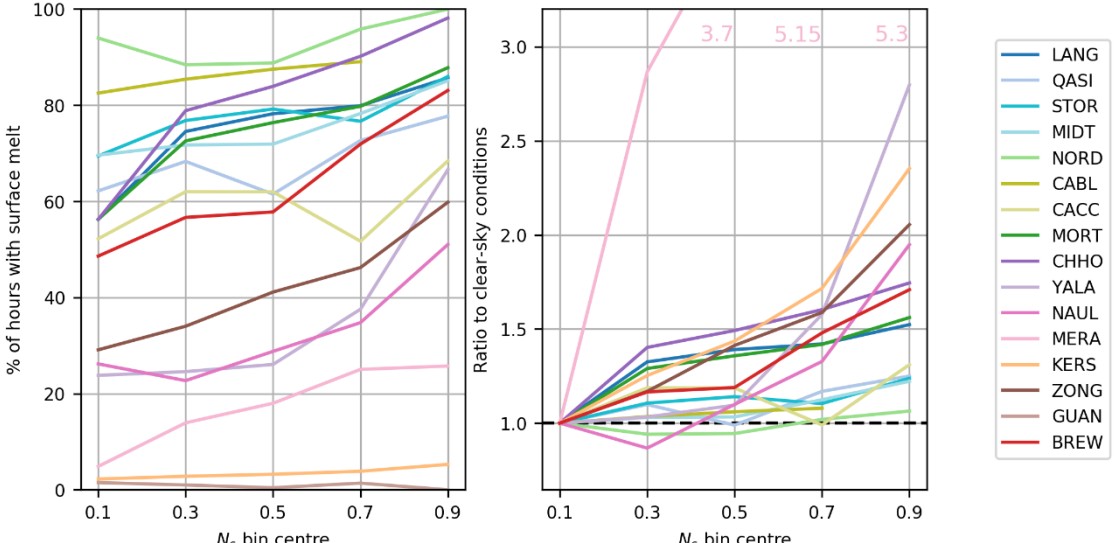

**Figure 9: (a) Percentage of hours with surface melt for different cloud conditions ($N_\varepsilon$) during melt season, (b) as for (a) shown as fraction with respect to clear-sky conditions ($N_\varepsilon <=0.2$). Note GUAN is excluded from panel (b) due to insufficient datapoints and for clarity some points for MERA are shown as text within the panel.**

In contrast to the percentage of hours with surface melt, the relationship between the amount of energy available for melt ($Q_M$) and cloudiness does not show a universal variation, with sites showing increased, decreased or no change with increasing

cloudiness on average (Figure 10). Around half the sites show a general reduction of daily average $Q_M$ with increasing cloudiness, particularly those in North America (CABL, CACC, NORD) and some European sites (MIDT, MORT, STOR) along with QASI and CHHO. LANG, MERA and KERS show large relative increase in $Q_M$ with cloudiness, while BREW, ZONG and YALA show a more mixed response with a small increase in melt in overcast conditions. LANG and NAUL display a sharp change from clear-sky conditions to the first partial cloud bin ($N_\varepsilon \sim 0.3$), but little change with increasing cloudiness.

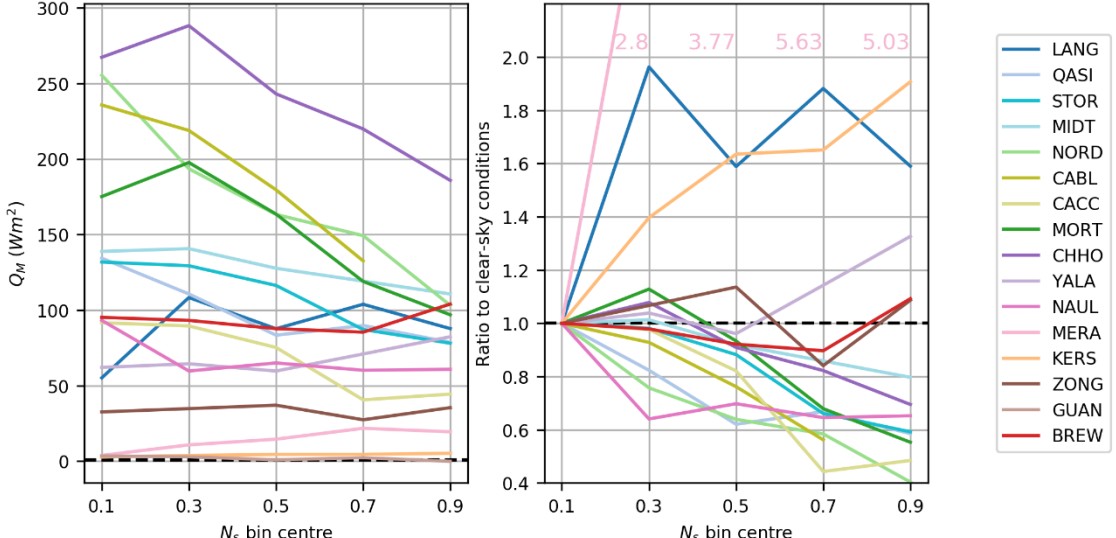

**Figure 10: (a) Average melt-season $Q_M$ for different cloud conditions ($N_\varepsilon$) (b) as for (a) shown as fraction with respect to clear-sky conditions ($N_\varepsilon <= 0.2$). Note GUAN is excluded from panel (b) due to insufficient datapoints and for clarity some points for MERA are shown as text within the panel.**

As cloudiness increases, the source of $Q_M$ changes; at all sites, the contribution of *SWnet* reduces and a greater proportion of $Q_M$ comes from the temperature-dependent fluxes (*LWnet*, $Q_S$ and $Q_L$) (Figure 11a,f; see Figure A5 for absolute values). At almost all sites, *LWnet* changes sign with cloudiness, from an energy sink in clear sky to an energy source in overcast conditions. At colder and drier sites (KERS, MERA, GUAN, NAUL, YALA, ZONG), negative $Q_L$ reduces $Q_M$ during clear-sky periods, but this effect reduces towards 0 as cloudiness increases. At the coldest sites (KERS, MERA and ZONG), $Q_L$ remains negative during melt (indicating evaporation as $T_s = 0\ °C$) even in overcast conditions. At BREW and CHHO, $Q_L$ switches sign with cloudiness, from an energy sink during clear-sky condition to an energy source in overcast conditions, while other mid and high latitude sites show modest increases in $Q_L$ with cloudiness. Small $Q_S$ fluxes at MERA, NAUL, YALA, ZONG are due to $T_a$ values during melt remaining around 0 °C.  At other sites, the proportion of melt from $Q_S$ remains fairly static with cloudiness, despite decreasing in absolute magnitude (Figure A5) due to decreases in $T_a$ (Figure 8a). The exceptions are BREW, MIDT, and QASI where the contribution from $Q_S$ increases with cloudiness and ZONG where the contribution of

$Q_S$ decreases. Note that as Figure 11 presents averages for only periods with surface melt, *LWout* is constant and changes in *LWnet* are entirely due to *LWin*.

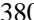

**Figure 11: Average melt-season SEB terms during hours with surface melt for different cloud conditions ($N_\varepsilon$). Variables are shown as a fraction of average $Q_M$ during hours with surface melt in each respective cloud condition ($N_\varepsilon$). Note y-axis range differs between panels.**

### 3.5 Relationships between $Q_M$ cloud effect and site characteristics

While the average change in $Q_M$ with cloudiness is small at some sites, it is instructive to assess whether the melt-season average $Q_M$ cloud effect (CE) at the various sites can be related to geographic or climatic parameters. Figure 12a,b shows the relationship between average cloudiness and melt at the various sites does not directly relate to latitude or altitude. Average near-surface air temperature is moderately correlated to $Q_M$ CE (Figure 12c). Sites with lower $T_a$ (e.g. MERA, KERS) generally

have smaller $Q_M$ CE than sites with higher high $T_a$ (NORD, CABL, MORT), but with some notable exceptions (e.g. LANG has positive $Q_M$ CE with relatively high $T_a$). Average cloudiness shows some association to $Q_M$ CE with clearer sites tending to have more negative $Q_M$ CE (Figure 12e), with the exception of tropical/arid sites with predominately clear-skies (KERS, GUAN) that show neutral $Q_M$ CE. Neither, average wind speed or relative humidity show a clear relationships with the $Q_M$ CE (Figure 12d,f). Average turbulent heat fluxes and *LWin* are moderately correlated $Q_M$ CE (Figure 12g,h,j), largely following the pattern of sites shown for $T_a$, while average *SWin* is not significantly correlated (Figure 12i).

Considering the association of radiative and melt cloud effects, average incoming radiation cloud effects explain some of the variance of $Q_M$ CE, with *LWin* (Figure 12l) showing a stronger association than *SWin* CE (Figure 12k). Combined, the incoming radiation cloud effects can explain over half (53%) of the variation in $Q_M$ CE (Figure 12m). Surface albedo has a similar correlation to $Q_M$ CE (Figure 12n) as the incoming radiation cloud effects together. The combination of these into the *Rnet* CE shows the clearest relationship to $Q_M$ CE (Figure 12o). In general, sites that experience a radiation paradox (LANG, ZONG, MERA) also experience greater melt in cloudy conditions (positive $Q_M$ CE), while sites with negative *Rnet* CE experience less melt in cloudy conditions (Figure 12o).

Turbulent flux cloud effects are also moderately correlated to melt-season average $Q_M$ CE (Figure 12p,q) and when combined explain approximately 44% of the variance in $Q_M$ CE (Figure 12r). Thus, sites where $Q_S$ decreases with cloudiness show more negative $Q_M$ CE. Sites where $Q_S$ varies little with cloudiness and/or $Q_L$ becomes less negative/more positive during cloudy periods show neutral or positive $Q_M$ CE. Interestingly, the radiative and turbulent heat cloud effects show a moderate association, with sites with large negative *Rnet* CE also having a negative net turbulent flux cloud effect, and vice versa (Figure 12s).

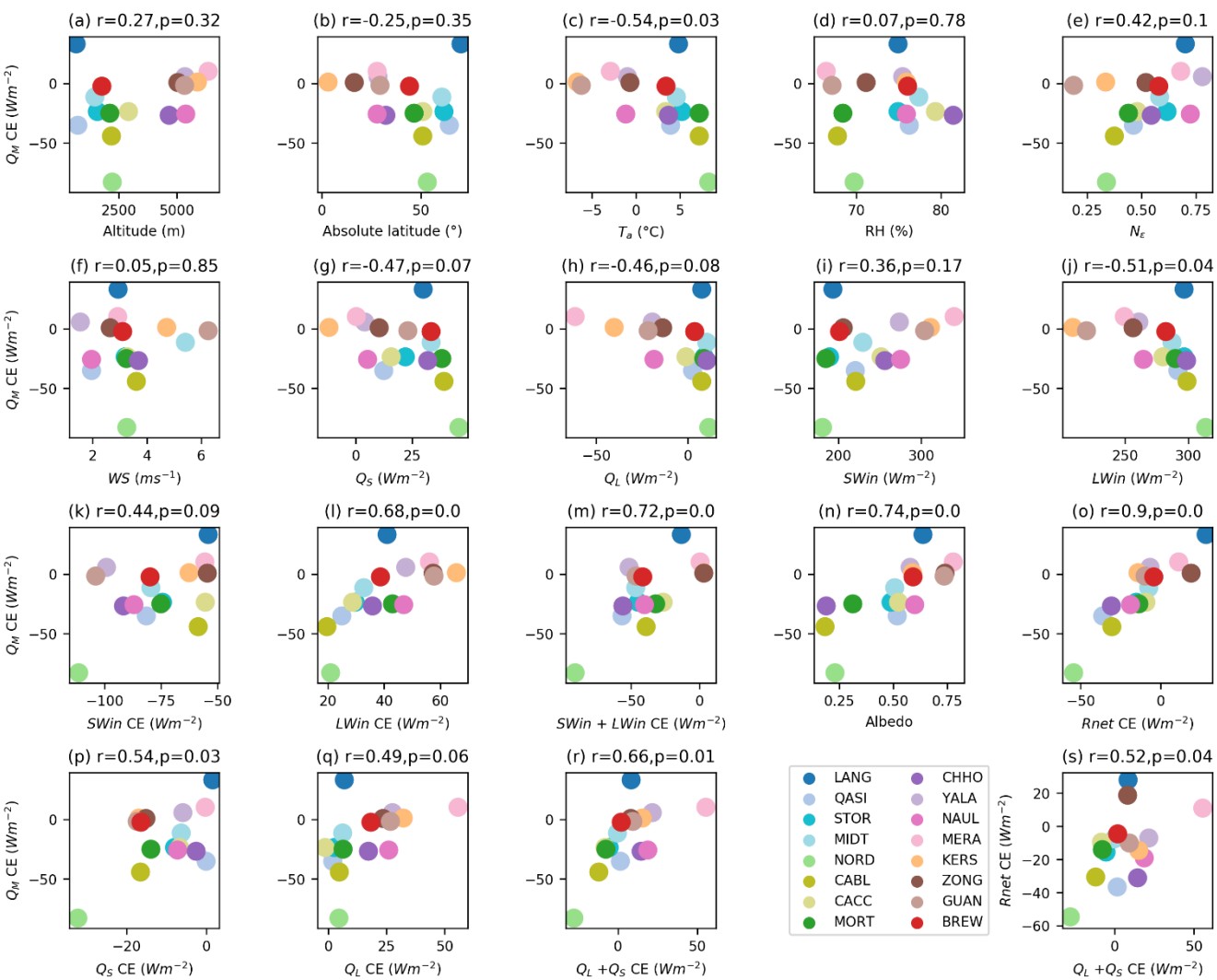

**Figure 12: The variation of average melt-season $Q_M$ cloud effect (CE) with (a) station altitude, (b) absolute station latitude, average melt-season (c) $T_a$, (d) $RH$, (e) $N_\varepsilon$, (f) wind speed ($WS$), (g) $Q_S$, (h) $Q_L$, (i) $SWin$, (j) $LWin$, (k) $SWin$ CE, (l) $LWin$ CE, (m) $SWin+LWin$ CE, (n) albedo, (o) $Rnet$ CE, (p) $Q_S$ CE, (q) $Q_L$ CE, (r) $Q_{s+L}$ CE. (s) is variation of $Rnet$ CE with $Q_{s+L}$ CE. See Section 2.5 for definition of CE. Average melt-season values are calculated by averaging values from the 5 cloudiness bins equally.**

# 4 Discussion

## 4.1 Regional and elevational patterns

Two groups of sites with a broadly similar response emerge from the above analyses, largely split by latitude, but also air temperature and continentality. The first group (YALA, NAUL, MERA, KERS, ZONG) consists of high-altitude sites in tropical regions and the Himalaya (excluding CHHO). These sites are comparatively cold, with negative $Q_L$ and small $Q_S$ during melt (Figure A5d,e). During cloudy conditions, these sites experience warmer and calmer conditions (Figure 8a,b), reduced evaporation/sublimation (less negative or, at times, positive $Q_L$; Figure A5e) and a large increase in the fraction of

time that melt occurs (Figure 9), regardless of the seasonality of cloud or the typical cloud conditions (e.g. KERS vs MERA). These sites also generally experience greater $Q_M$ in cloudy periods (except for NAUL; Figure 10) when averaged over a long melt season that includes months with marginal melt conditions. Some sites experience a radiation paradox where *Rnet* increases with cloudiness, while others show a small decrease in *Rnet* with cloudiness (Figure 7f). While GUAN experiences similar patterns of near-surface meteorology and radiation as the sites in this group, it experiences very infrequent melt (Figure

9a).

The second group consists of the mid- and high-latitude sites outside the Himalaya (LANG, QASI, STOR, MIDT, NORD, CABL, CACC, MORT, BREW) as well as CHHO. These sites experience higher average melt-season $T_a$, and $T_a$ generally decreases with cloudiness (Figure 8a). Despite decreased $T_a$, melt becomes more frequency in cloudy conditions (Figure 9).

With a few exceptions (e.g. BREW, LANG), $Q_M$ decreases with increased cloudiness, though the magnitude of decrease varies widely (from 20% to 60% less in overcast compared to clear-sky conditions; Figure 10). CHHO stands out from the other Himalayan sites in that it has a higher average $T_a$ that does not vary greatly with cloudiness (Figure 8a). Here also, low albedo drives a strong negative *Rnet* cloud effect (Figure 7f) that, in turn, drives a large decrease in $Q_M$ during cloudy periods (Figure 10). At all these sites, $Q_S$ is positive in all cloud conditions (Figure 11d), though the absolute magnitude is generally reduced

in cloudy periods due to decreased $T_a$ (Figure A5d). Cloud is associated with increased wind speed at most maritime sites (LANG, MIDT, STOR, BREW; Figure 8b) but does not show a consistent relationship to $Q_M$ (Figure 10); MIDT and STOR experience less $Q_M$ in cloud conditions, whereas LANG and BREW experience greater $Q_M$ due to increased wind speed and comparatively modest decreases in $T_a$ that drive increased *LWnet* and more positive $Q_L$ (Figure A5e). In the case of LANG, increased $Q_M$ during cloud is also due to a positive *Rnet* cloud effect (Figure 7f).

Locations with AWS at two elevations highlight more positive *Rnet* cloud effects at accumulation sites than ablation sites due to the higher albedo (Figure A1) and larger difference between clear-sky and overcast emissivity (Figure 4). Differences in melt are stronger at the Himalayan pair (NAUL, MERA), where melt is decreased in cloudy conditions at the lower sites and

increased during cloud at the upper site (Figure 10). At the pair in Canada (CABL, CACC), both sites experience reduced melt during cloudy conditions, though in absolute terms, the decrease is larger in the ablation area.

## 4.2 Limitations

The derivation of cloudiness from *LWin* also poses challenges. At some sites (e.g. LANG, and MORT), $\varepsilon_{cs}$ shows a poor fit at higher vapour pressure, with incoming *LWin* during clear-sky periods being higher than that expected from the theoretical curves (Figure 4). This mismatch between theoretical and observed $\varepsilon_{cs}$ during periods of higher $e_a$ may cause some clear-sky periods to be misclassified as being in the first partial cloud bin ($N_\varepsilon \sim 0.3$). Indeed, at both LANG and MORT, the $N_\varepsilon \sim 0.3$ bin shows higher melt, indicating this may be the case. The reasons for this mismatch have not been investigated, but it may be due to a different method used to correct *LWin* data (Giesen et al, 2014) or changes in water vapour profiles in the atmospheric boundary layer. There is also some unavoidable degree of circularity in analysing longwave radiation fluxes (Figures 6 and 7) that have also been used to derive cloudiness. However, as *LWin* does not solely depend on cloudiness, but also on variations in $T_a$ and *RH*, the circularity is not complete. For instance, at Brewster Glacier, the increase in *LWin* between clear-sky and overcast conditions is approximately the same as the change in clear-sky *LWin* due to seasonal variations in $T_a$. Because the method used to calculate cloudiness accounts for the effect of $T_a$ and *RH* on *LWin*, the effect of these variations in near-surface meteorology on *LWin* is retained in the analyses shown in Figures 6 and 7.

While efforts have been made to homogenise the datasets, it is possible that biases still affect the results. Interannual variability causes uncertainty, particularly for sites with only one or two seasons (e.g. NORD, ZONG). Giesen et al. (2008 Table 4) show that at MIDT, the contribution of SEB components to melt during clear-sky periods can vary up to 12% between years, while variability in overcast periods is less. The interannual variability is partly influenced by the seasonality of anomalies in cloudiness, with strong anomalies in spring causing the importance of $Q_S$ to melt to change markedly. Some sites also have discontinuous records (CABL, CACC, NORD, CHHO) that do not include periods with lower melt rate outside the peak melt season. Increased clear-sky solar radiation and $T_a$ as well as decreased albedo during the peak melt season are likely to cause *Rnet* and $Q_M$ cloud effects to be larger at these sites compared to those with longer records that include periods of more marginal melt. This effect is demonstrated by repeating the analysis but restricting the melt season to months with at least 80% of the maximum monthly-average $Q_M$, 2-3 months at each site (Figure A6). Figure 13 shows the relationship between average $Q_M$ and $N_\varepsilon$ for the period with peak melt rates at each site. The previously large increase in $Q_M$ with cloud at MERA and LANG becomes more variable, and $Q_M$ is smaller in overcast conditions compared to clear-sky. This is primarily due to the removal of months with a high albedo snow surface in the early season where a strong radiation paradox drives an increase in melt during cloud periods. In clear-sky conditions, higher $T_a$ and $e_a$ in the peak melt season creates generally positive $Q_L$ at these sites (not shown). BREW also now shows a moderate decrease in $Q_M$ with cloud, while ZONG shows a much stronger decrease due to marked seasonal changes in the SEB terms driving melt (less negative *LWnet* and $Q_L$ in austral spring and summer;

Figure A2). Only one site (YALA) still shows its highest $Q_M$ in overcast conditions, but the increase is small compared to the average for the longer melt season. In fact, at outer-tropical sites such as ZONG where melt can occur in most months alongside large seasonal variations in precipitation and cloudiness, the analysis here likely mixes cloud effects with seasonal changes of other meteorological forcings (such as potential solar irradiance, humidity and air temperature).

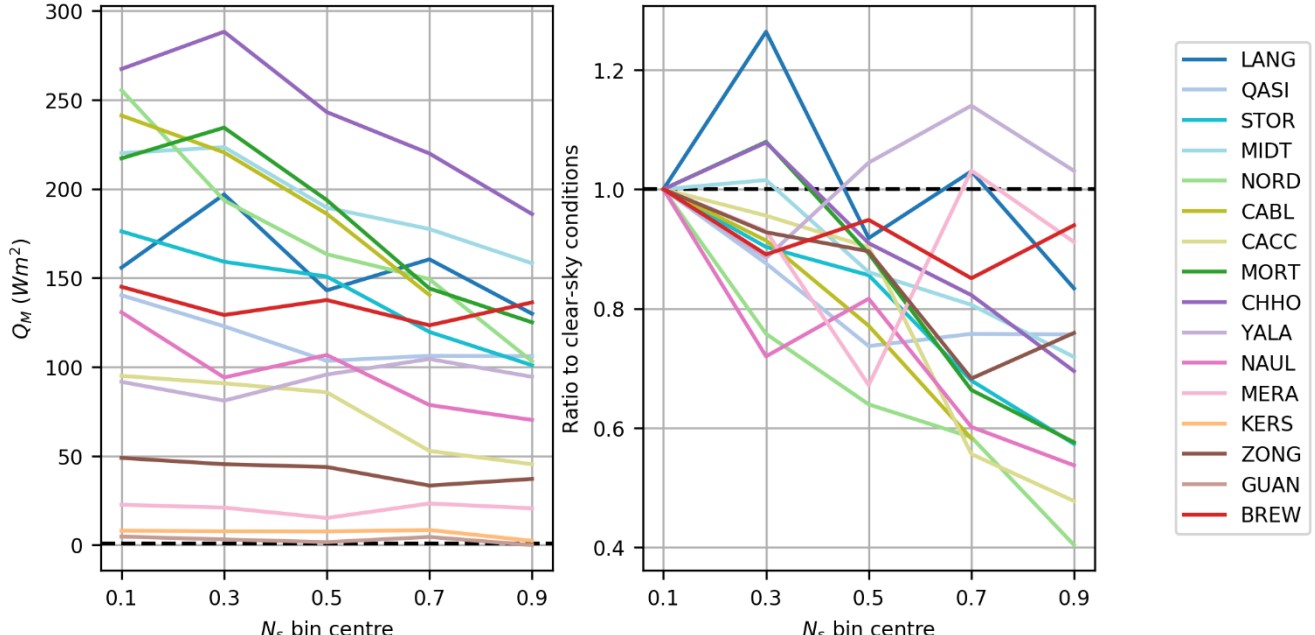

**Figure 13: As for Figure 10 but only for months with > 80% of maximum monthly-average $Q_M$. Note GUAN and KERS are excluded from panel (b) due to insufficient datapoints.**

Seasonal changes in cloud effects on melt have been previously reported by some studies; Giesen et al. (2008) show that negative $Q_M$ cloud effects at MIDT were restricted to July and August, with other months showing neutral or positive cloud effects; Conway and Cullen (2016) show only one month with negative $Q_M$ cloud effect at BREW, with positive effects in other months; Chen et al. (2021) report strong negative $Q_M$ cloud effects in July and August for Laohugou Glacier No. 12 in the western Qilian Mountains of China, with weaker negative effects in May and June, and neutral effects in September. To elucidate spatial patterns of net melt cloud effect, future studies should investigate seasonal patterns of cloud effects, and establish the timing of transitions between periods of positive and negative $Q_M$ CE and how these relate to *Rnet* CE and surface meteorology. It is likely that the timing of transition from positive to negative $Q_M$ CE will therefore determine the melt-season average cloud effects., To this end, there is a need to capture AWS records through the full annual cycle at study sites in order to fully understand the relationships between meteorological forcing and melt.

### 4.3 Mechanisms influencing SEB changes with cloud

In addition to the key role that surface albedo plays in determining $Rnet$, there are three key mechanisms that drive temporal changes in SEB with cloudiness

i)            direct forcing of incoming radiation (decreased $SWin$ and increased $LWin$),

          ii)          changes to near-surface meteorology that alter turbulent heat fluxes

          iii)         surface and subsurface temperature feedbacks that alter net radiative and turbulent fluxes

Here we demonstrate that direct forcing of incoming radiation and surface albedo explains much of the net effect of clouds on
$Q_M$ across sites. The high correlation between melt-season average $Rnet$ CE and $Q_M$ CE between sites (Figure 12), along with the sensitivity of these averages to the length of the melt season (Figure 10 vs Figure 12) underlines the primary control of direct and indirect radiative mechanisms on determining the sign of melt response to cloud. It is likely that substantial seasonal variations of $Rnet$ CE exert the primary control on the effect of clouds on glacier melt.

Changes in turbulent heat flues with cloudiness tend to be smaller in magnitude than changes in $Rnet$ (Figure A5), except for the more extreme cases where air temperature changes greatly with cloudiness, e.g. NORD, where $Q_S$ markedly decreases with cloud and MERA, where $Q_L$ becomes far less negative during cloud. Despite this, net turbulent heat flux cloud effects show moderate correlation to $Q_M$ CE, and thus changes in near-surface meteorology play a significant role in determining the net response of melt to cloud. These findings echo those of Liu et al. (2021) who show increased melting during cloudy periods
on Mt Everest are due to increased $Rnet$ as well as lower wind speeds that drive smaller losses to $Q_L$, and Conway et al. (2016) who found changes to $Q_L$ contributed to increased melt during cloudy periods. Future work should assess the mechanisms driving the observed covariance between cloudiness and near-surface meteorology at different sites, e.g. Do large-scale changes in airmass or local/meso-scale processes drive changes in $T_a$ with cloud? How well are these processes represented in the datasets used to force glacier melt models on regional scales? Seasonal changes in the relative magnitudes of turbulent
and radiative cloud effects also deserve further scrutiny.

Surface temperature responds quickly to changes in SEB, and here we show that during cloudy periods, a melting state is observed more frequently, in line with previous research on maritime glaciers (Conway et al., 2016). We have not attempted to analyse further surface and sub-surface temperature feedbacks here as not all datasets contain these variables and a detailed
analysis is more suited to sensitivity experiments that allow the transient response of sub-surface temperature, humidity and refreezing to be resolved.

The increased frequency of melt during cloudy conditions, especially at higher elevations, raises the question of how glacier-wide melt is altered by clouds, along with how glacier-wide surface mass balance is altered by refreezing. Van Tricht et al.,

(2016) show increased runoff from the Greenland Ice Sheet during cloudy periods due to increased melt extent and decreased refreezing of melt water, while Niwano et al. (2019) found clouds increase melt extent but reduce total melt due to feedbacks between cloudiness and near-surface humidity. These studies are in line with the findings here – that clouds enhance the possibility of melt at a given site, by removing large negative *LWnet* and $Q_L$ fluxes to precondition the surface to melt, but do not necessarily cause greater melt unless albedo is high enough to cause a radiation paradox or unless increased near-surface

air temperature, humidity and/or wind speed causes an increase in net turbulent fluxes.

## 4.4 Implications for glacier melt modelling

Previous research that identified a higher sensitivity to warming associated with cloud at BREW (Conway and Cullen, 2016), showed this occurred without increased melt during cloud periods. The effect was primarily due to increased melt frequency and temperature-dependent fluxes during cloudy periods as well as accumulation-albedo feedbacks. All sites analysed here

show increased melt frequency and temperature-dependent fluxes during cloudy periods, suggesting more sites may also experience a higher sensitivity to warming associated with cloud. While a formal analysis is beyond the scope of this paper, we may therefore expect that the response of melt to past and future temperature change will be modified by changes to atmospheric moisture in the form of clouds and vapour fluxes. The simplified temperature-index models that are generally used to predict future glacier change on global and regional levels (e.g. Marzeion et al., 2018; Huss and Hock, 2018; Zekollari

et al., 2019) do not account for these effects. Enhanced temperature-index models that can account for changes in cloudiness through solar radiation (e.g. Pellicciotti et al., 2005) only include the opposite effect – a reduction in solar radiation by clouds – and therefore may underestimate future melt at sites where cloud cover is not universally associated with reduced melt (e.g. high altitude and maritime glacier sites). Given the positive effect of clouds on net radiation at snow covered and high-altitude sites, future increases in cloud cover may promote further melt, especially during marginal melt seasons and especially at high

elevations. However, caution is warranted in making generalisations as the analysis here shows that even in this set of 16 glaciers, we find variability in the links between clouds and melt, and it seems that some processes are site specific even in this small sample.

The non-linear relationships between clouds and melt motivates the use of SEB models in regional and global assessments of

glacier response to climate change. To aid in the development of globally and regionally applicable SEB models and parameter sets, the research community should investigate creating a central open-source repository for glacier AWS and SEB datasets along with supporting metadata. Such a repository would facilitate the easy transfer of data between researchers, streamline processing by establishing data format and metadata standards, as well as motivating best-practice in data collection and quality control. Alongside this, careful assessments of *SWin* and *LWin* and their relationship to near-surface meteorology from global,

regional and meso-scale meteorological models should be undertaken to ensure uncertainties in model input data are reduced and to assess the need for downscaling to account for local-scale processes. As many glacier SEB models rely on empirical

relationships between *SWin* and *LWin* to modify these variables to account for local-scale changes in near-surface meteorology (e.g. Mölg et al., 2009a; Conway et al., 2015), globally applicable parameterisations of *SWin* and *LWin* should be tested.

**Conclusions**

Sixteen high-quality published datasets of near-surface meteorology, radiation, and surface energy balance over glaciers in very different climate settings have been homogenised and analysed in a common framework. The analyses sought to assess how the relationships between clouds, near-surface meteorology and surface energy balance vary in different mountain glacier environments. Distinct regional differences in the seasonality of cloudiness are demonstrated between different mountain glacier environments. On average, over the main period of melt at each site:

- Near-surface humidity (both relative and absolute) is shown to universally increase in cloudy conditions. In contrast, a divergent relationship is found between near-surface air temperature and cloudiness; at colder sites (average near-surface air temperature in melt season < 0 °C), air temperature is increased in cloudy conditions, while for warmer sites (average near-surface air temperature in melt season >> 0 °C), air temperature decreases in cloudy conditions. In essence, air temperature tends towards the melting point of ice in cloudy conditions. Wind speed shows a mixed
association to cloudiness at different sites.

  - Most sites, on average, show a modest to strong decrease in net radiation during cloudy conditions during the melt season. A few sites show a clear increase in net radiation with cloud – aka 'radiation paradox' – but this result is sensitive to the months used in the analysis due to seasonal changes in incoming radiation fluxes and albedo.

  - At all sites, surface melt is more frequent in cloudy compared to clear-sky conditions.

- At all sites, temperature-dependent fluxes contribute a larger fraction of melt energy during cloudy conditions, primarily due to increased incoming longwave radiation and less negative and/or more positive turbulent latent heat fluxes. The contribution of turbulent sensible heat generally varies little with cloudiness.

  - Cloud cover does not affect daily total melt in a universal way; some sites show average melt energy increases in cloudy conditions while at other sites, average melt energy decreases. The complex association of clouds and melt
energy is due to the interaction of multiple physical processes (direct radiative forcing, surface albedo, co-variance with temperature, humidity, and wind) that force it to vary widely with latitude, average melt-season air temperature, degree of continentality, season, and elevation. Overall, the association of clouds and melt is most closely related to net radiation cloud effect, with sites displaying a radiation paradox also showing an increase in energy for melt in cloudy conditions.

- It is likely that substantial seasonal variations in *Rnet* CE exert the primary control on the effect of clouds on glacier melt, through changes in surface albedo and the balance of incoming radiation fluxes. Changes in net turbulent fluxes also play a role, and the mechanisms driving co-variance between clouds and near-surface air temperature, humidity and wind speed should be more widely explored.

The non-linear relationships between clouds, near-surface meteorology and melt motivate the use of physics-based surface energy balance models for understanding future glacier response to climate change, particularly in areas where atmospheric moisture plays a key role both in accumulation and ablation processes (e.g. Himalaya, tropical glaciers, maritime glaciers). Future work should also look to carefully assess shortwave and longwave radiation fluxes and their relationships with near-surface meteorology in global, regional and meso-scale meteorological model analyses if we are to confidently use these tools to better understand how future glacier melt will respond to changes in atmospheric temperature.

**Data and code availability**

AWS data is available from individual paper authors listed in Table 1. Analysis code can be accessed at https://github.com/jonoconway/cloud-glacier.

**Author contributions**

JC conceptualized the study, curated the data, conducted the formal analyses, and wrote the manuscript. Other co-authors supplied data suitable for curation, aided in the investigation and reviewed/edited the manuscript.

**Acknowledgements**

JCs contribution to this research was supported by the Royal Society of New Zealand Marsden Fund. The authors wish to thank the following additional people and organisations for data contributions and funding data collection and processing: Maxime Litt, Hans Oerlemans, GLACIOCLIM (UGA-OSUG, CNRS-INSU, IRD, IPEV, INRAE), International Joint Laboratory LMI GREAT-ICE (IRD, EPN-Quito), PROMICE, the Greenland Ecosystem Monitoring Programme, ICIMOD, Antoine Rabatel (IGE), Alvaro Soruco (UMSA, Bolivia), NSERC Discovery Grant and Research Tools and Instruments, Canada Foundation for Innovation. MFA acknowledges research grants from IFCPAR and IRD, France. Patrick Wagon is thanked for his helpful comments on the manuscript. We also thank the editor and two anonymous reviewers for their comments, which have improved the final manuscript.

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

**Appendix:**

**Table A1: Optimised clear-sky emissivity coefficients and error in $\varepsilon_{cs}$.**

| Site | Fitted value of $b$ | Root-mean squares error of calculated $\varepsilon_{cs}$ vs $\varepsilon_{eff}$ in selected clear-sky conditions |
|:---:|:---:|:---:|
| BREW | 0.443 | 0.0190 |
| CHHO | 0.538 | 0.0280 |
| CABL | 0.483 | 0.0199 |
| CACC | 0.436 | 0.0190 |
| GUAN | 0.379 | 0.0292 |
| KERS | 0.291 | 0.0236 |
| LANG | 0.458 | 0.0201 |
| MERA | 0.472 | 0.0391 |
| MIDT | 0.428 | 0.0166 |
| MORT | 0.398 | 0.0240 |
| NAUL | 0.495 | 0.0378 |
| NORD | 0.489 | 0.0202 |
| QASI | 0.466 | 0.0124 |
| STOR | 0.463 | 0.0171 |
| YALA | 0.468 | 0.0240 |
| ZONG | 0.443 | 0.0251 |

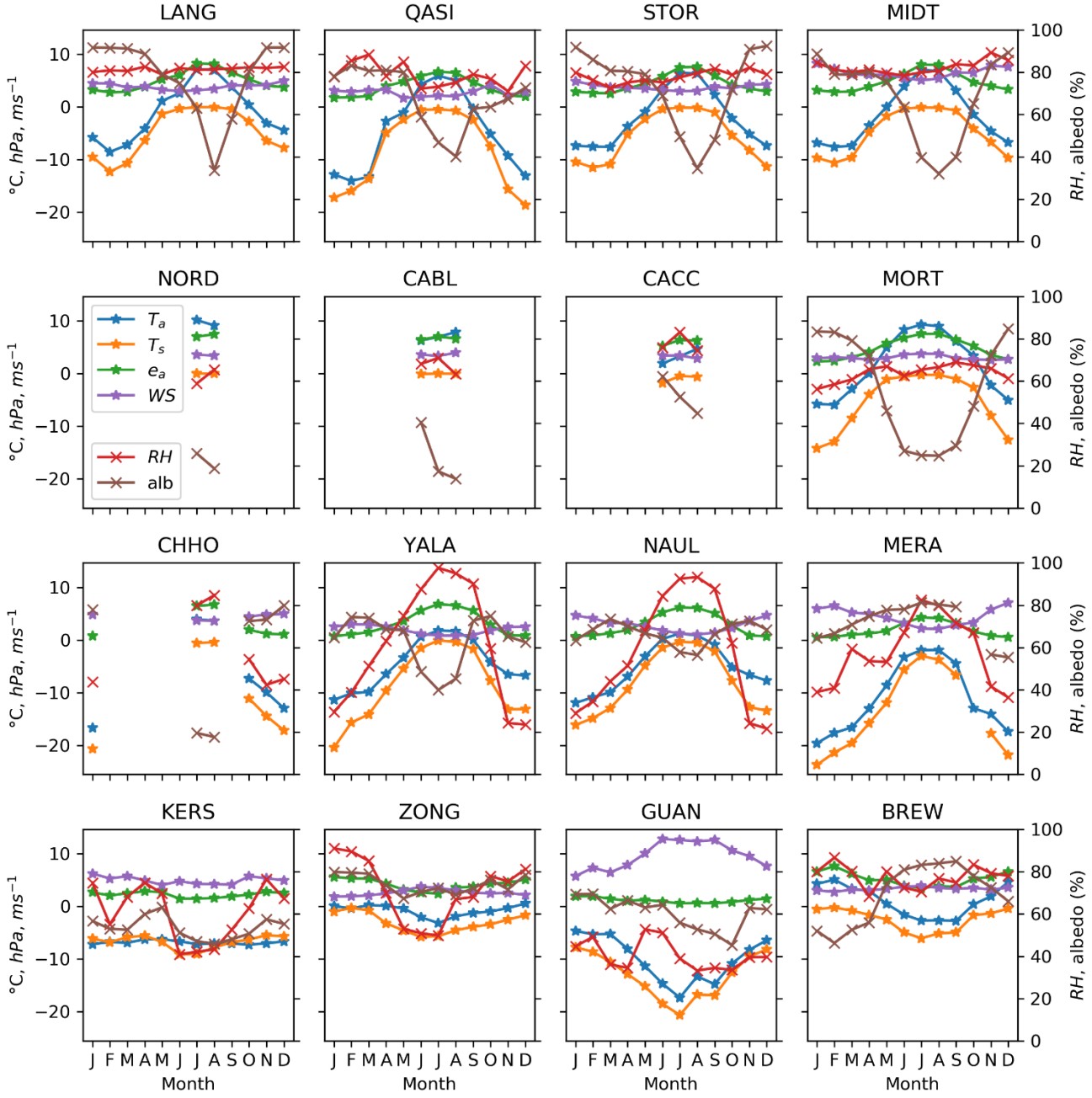


**Figure A1: Monthly average near-surface meteorological conditions at each site. Note monthly value only shown for a site if > 10 complete days in month across full record.**

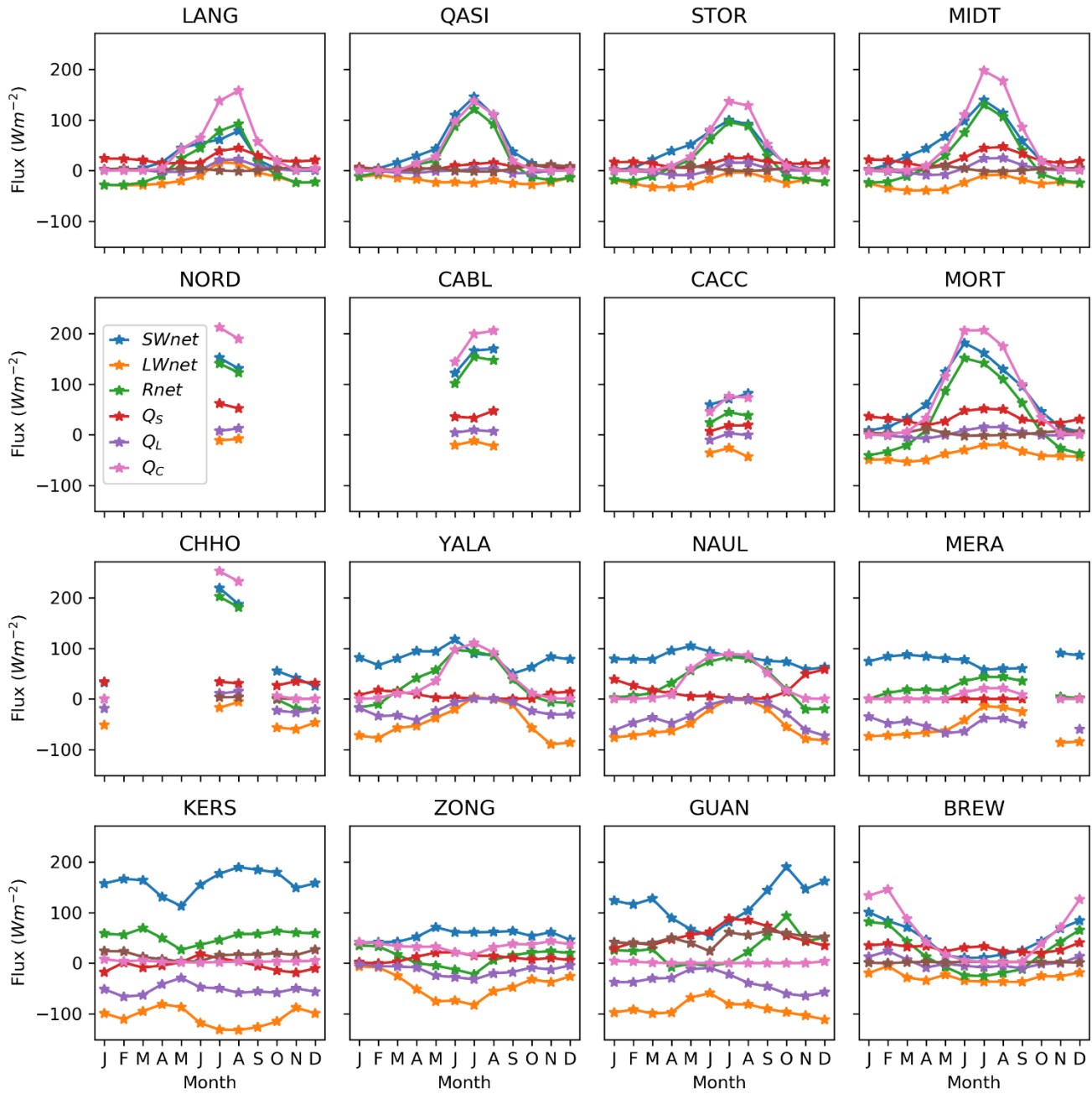

**Figure A2: Monthly average SEB fluxes at each site. Note monthly value only shown for a site if > 10 complete days in month across full record.**

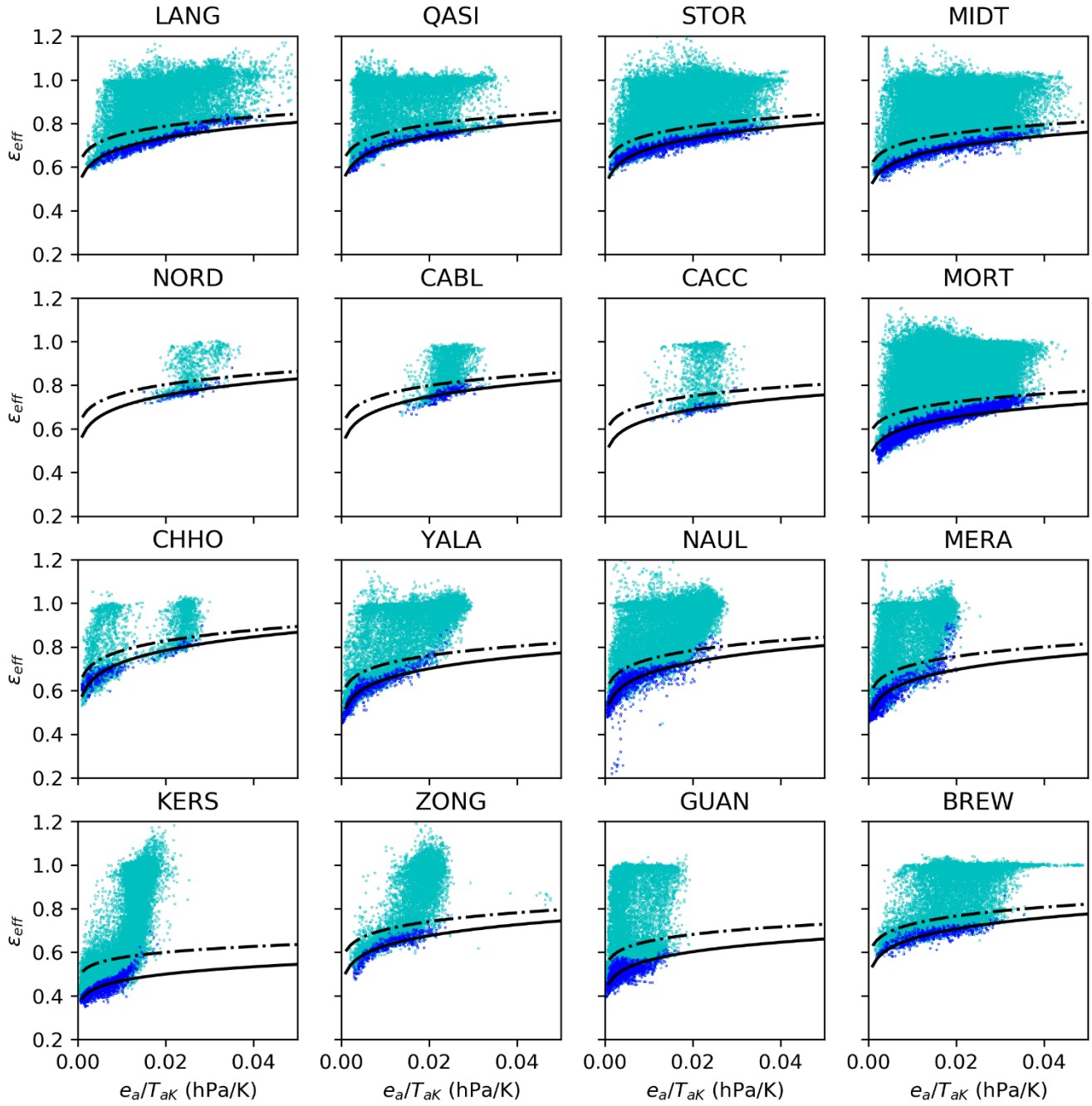

**Figure A3: Observed $\varepsilon_{eff}$ (points) and calculated $\varepsilon_{cs}$ (solid line) fitted to lowest 10% of *LWin* in 30 $e_a/T_{a.K}$ bins (shown in blue).**
**Calculated $\varepsilon_{eff}$ at clear-sky limit of $N_\varepsilon = 0.2$ (dash-dotted line).**

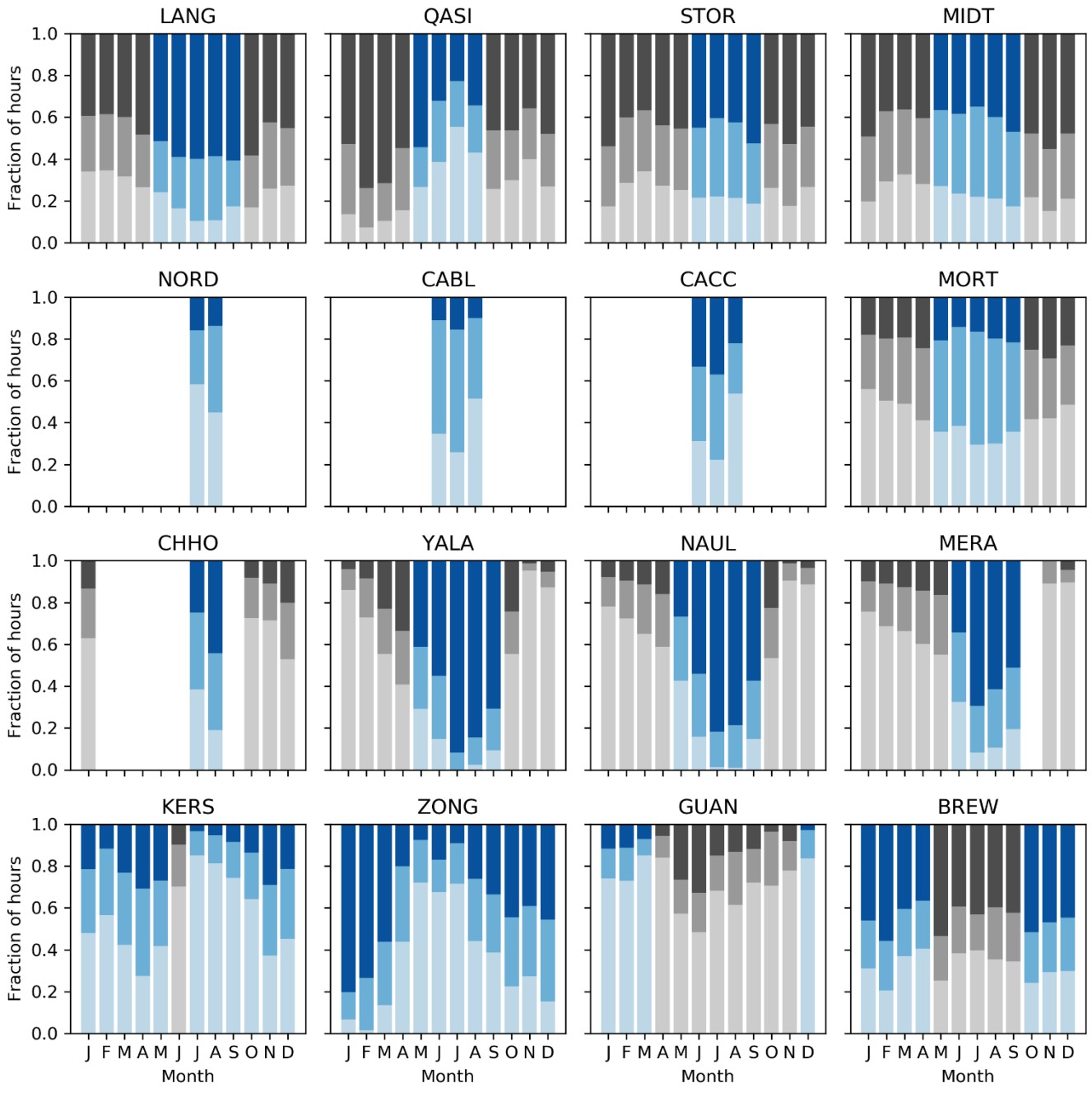

**Figure A4: Monthly fraction of clear-sky (light shading), partial-cloud (mid shading) and overcast conditions (dark shading) defined using hourly cloudiness ($N_\varepsilon$). Months defined as within the 'melt season' are shaded blue.**

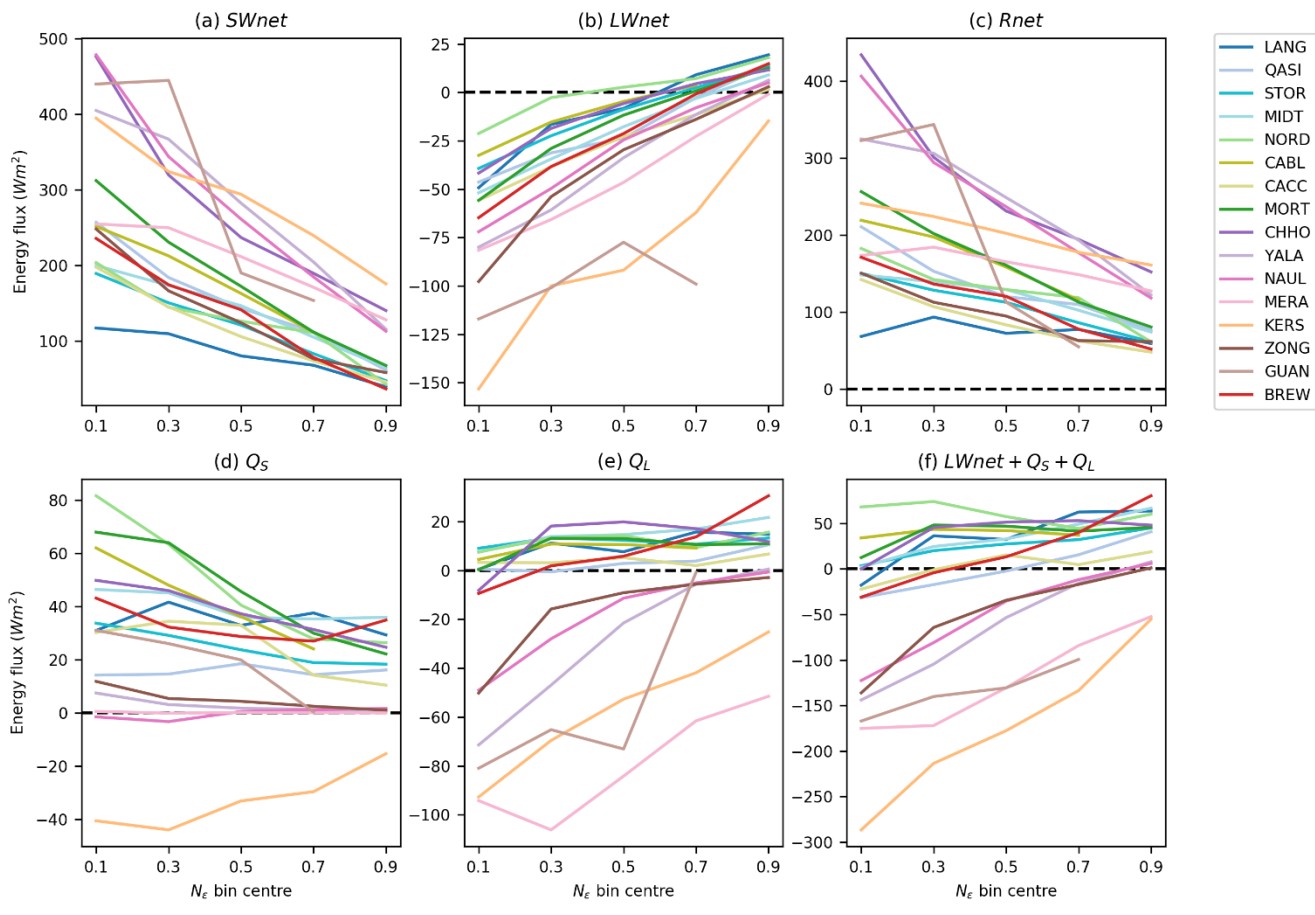


**Figure A5: Average melt season SEB terms during hours with surface melt for different cloud conditions ($N_\varepsilon$).**

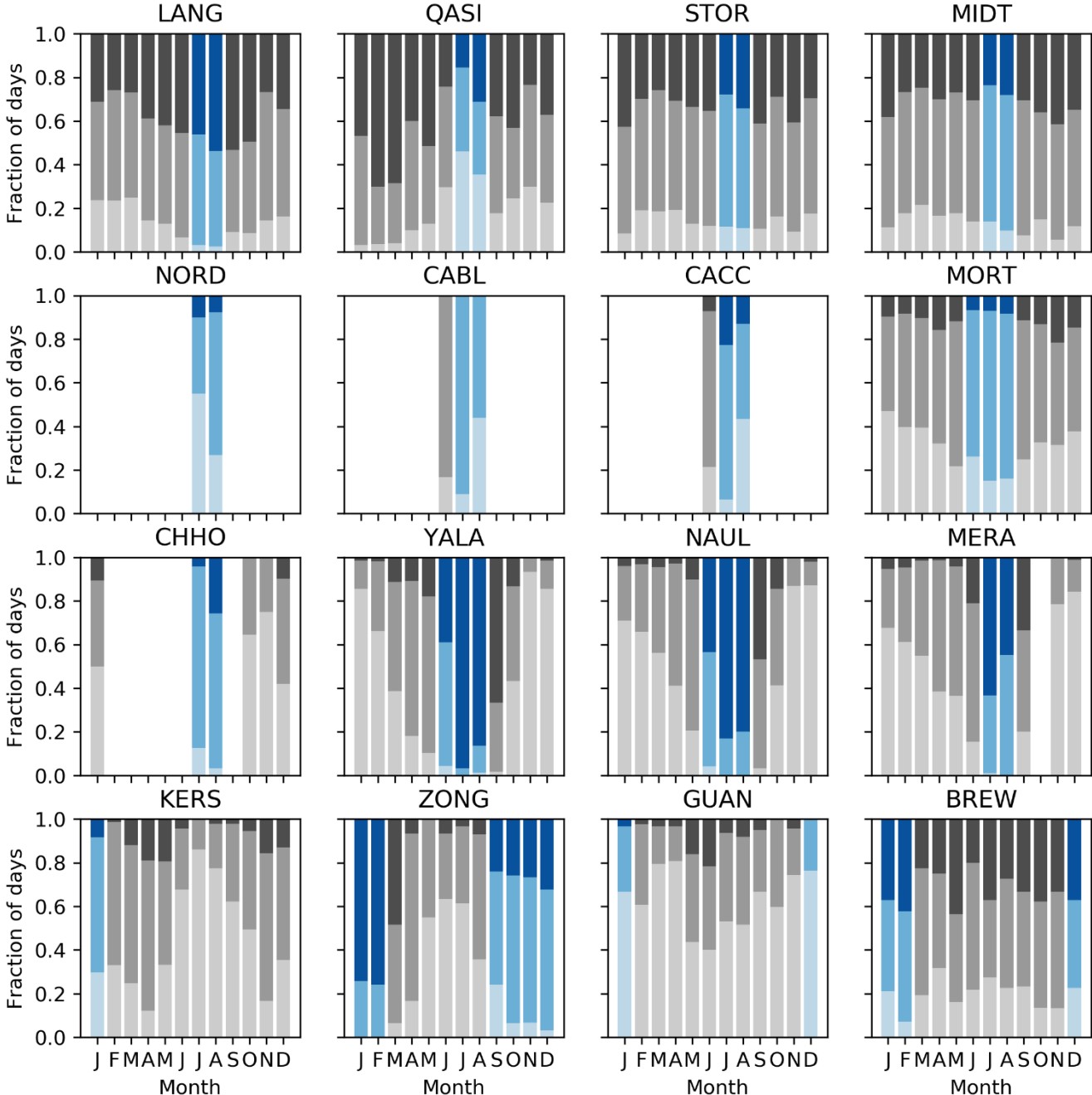

**Figure A6: As for Figure 5 but with months with > 80% of maximum monthly-average $Q_M$ shaded blue. Bars show monthly fraction of clear-sky (light shading), partial-cloud (mid shading) and overcast conditions (dark shading) defined using daily average cloudiness ($N_\varepsilon$).**