# Peer review of "Cloud forcing of surface energy balance from *in-situ* measurements in diverse mountain glacier environments"

_The Cryosphere, 2022_

## Referee Comment (RC2)

**Review of Conway et al. Cloud forcing of surface energy balance from in-situ measurements in diverse mountain glacier environments**

Conway et al. use a global selection of on-glacier AWS data to determine the effect of clouds on the surface energy balance. They investigate the influence of clouds on the near-surface meteorology, individual energy fluxes and the frequency and magnitude of melt. They found an increase in the frequency of melt during cloudy conditions but the effect of clouds on the energy available for melt varied spatially.

Overall, I think the purpose of the paper is a very good one, and it is certainly an interesting approach to look at the impacts of clouds across a range of sites, since often energy balance studies are confined to one site or a region, so the global aspect is appealing. The paper is also clearly written throughout, and the methodology followed is sensible. However, I do have a couple more significant concerns which I highlight in the major comments below:

**Major comments**

1. Depth of analysis and understanding of global trends

In the main results section the variation in the results (in terms of the cloud effect on the meteorology, surface fluxes and melt) caused by the different location, climate and elevation of the sites is mentioned (certainly when the effect is quite clear). However, I don't think the authors really make the best use of their dataset to fully interrogate the spatial variation in the results. The relationship between the cloud effect and station and energy balance characteristics is not investigated fully (with scatter graphs) until section 4.2 (which anyway should be a result section). Although the figures earlier in the results section are clear as they are they are not well suited to investigating the spatial differences and improving these figures to make the station characteristics clearer and including scatter graphs earlier in the results would be a good idea. Furthermore, the analysis in section 4.2 is not robust, the authors need to calculate the correlation and regression (if appropriate) coefficients and report them in the paper. Currently assessments are made only on a visual assessment. In general, I think the paper needs an extra stage of analysis yet to give its findings more credibility, and this may also allow clearer findings on why the effect of clouds on melt energy varies in sign spatially.

2. Discussion

The first two sections of the discussion are really still results, and then there are only the limitations and implications sections, with the latter section really missing references. I miss here a proper in-depth discussion which brings together the understanding alongside other studies. Consider answering some questions, e.g. Why do clouds influence the near surface meteorology in the ways you found? (explain the physical mechanisms) What factors cause changes in the impact of clouds on melt? You touch on this in sections 4.1 and 4.2 and also in your limitations, but I feel you need to set this out more as a clear discussion with each of the factors and their influence. There is also a need to bring in other energy balance studies which have looked at the cloud effect (or even seasonal differences where cloud cover likely varies markedly), especially in regions not covered by your analysis.

**Minor comments**

L56: To me the subsurface is more the material under the glacier, I think $Q_c$ is the conductive heat flux into the surface.

L57: Are all the fluxes in W m$^{-2}$ (they usually are but just state this)?

L86-88: It might be worth expanding on some of these methods to derive cloudiness and their main assumptions/difficulties.

L99: Define AWS here.

In methods: Maybe it would be useful to have a simple idea of the climate type/seasonality of each site? It seems that the overall broader climatology will drive the differences in the cloudiness patterns and how they relate to the melt seasons, so having this context early on would be useful.

L110: Due to the different methods of the calculation of the turbulent fluxes consider including a table with how the non-radiative fluxes were calculated for each site (in the appendix/SI would be fine).

Figure 1: Maybe include insets where you have several sites relatively close by in a region. Also use a different symbol to label colour for readability.

Sections 2.3, 2.4 and 2.5 – if these are all within 'data processing' then it might make sense for these sections to be 2.2.1, 2.2.2, 2.2.3 (so sub-sections of 2.2)

Figure 3 caption: 'Steps'

L149-162: Honestly, I think this could go into an appendix or SI. But I am wondering, if you had to do these quality checks then how do you know that the SEB fluxes were also calculated post these quality checks - I thought you were using the published data (which hopefully would already be checked?) Can you clarify this please. Calculating Ts from LWout works quite well but not always, give the reference for how you did this.

L172: Also define sigma as the Stefan Boltzmann constant, and give a reference for this equation.

Section 2.4 could probably be shortened.

Figure A2, A4: Please add a legend so the reader knows that the colours are used to define the melt season.

L224: Why only look at the cloud effects versus the radiative fluxes? You do look in Figure 11 at the importance the turbulent fluxes for melt and how that varies with cloudiness, but why not also include them as for the radiative fluxes in Figure 7. Furthermore, is there not some circularity in looking at the LWnet differences given that LWin is used to calculate the cloudiness?

Figure 4 caption: It would be useful to have a legend, or at least to explain what the darker via lighter colours represent. From the text it seems like darker colours = greater frequency of conditions, but this should also be clear from the figure/legend on its own. Consider outer boxes (or other methods) showing the splits between regions.

L258: 'between monsoon and arid regions *although it still shows an increase in partially cloudy conditions in the melt season*' Or something similar, just for clarity.

Figure 5: This is nice way to show the cloudiness at the sites, but it might be useful to have some overall metrics so its slightly quicker to compare sites, e.g. the mean and range of melt season monthly cloudiness? It might also be useful to group by region (Himalayan, European etc.) Even though I know most of these sites and where they are its not so easy to see trends, and I imagine it would be harder if you didn't know inherently the site locations.

Figure 6: Maybe it would be helpful to take this a step further, for instance can you relate the gradients of these lines to the site lat/long/elevation, e.g. in a scatter graph? It's easy to see that KERS, MERA and ZONG are different but harder to know what is causing the variation in the other sites. You do attempt this in the discussion but I think this analysis could be more thorough and come earlier in the paper.

L289: 'cloud effect is small and negative' It would be useful also to scatter this overall change in Rnet against the sites to see if there are regional clusters.

L293: 'more positive response to' - do you mean in terms of an increase in Rnet here?

L307: 'relationship is weak and non-linear' - Quantifying the strengths of these relationships and their gradients (for all the variables in Figure 8) would be a good idea.

L323: 'at all study sites' - this doesn't appear to be the case for GUAN.

L329 – 331 'While…..day and night' – Add a reference to this effect if you don't show the analysis yourself.

L331- 334: This sentence might be better earlier in the paragraph.

Figure 9: Here and in other similar figures, it might be a good idea to use different line styles as well as colours to indicate different regions? It might also allow you to use fewer colours, and have a palette which is more colour blind friendly. The strength of this paper is the wide range of sites and yet you need to show better the regional/climate/elevation differences in your plots.

L337: 'indicating *sublimation*' Since we are going from ice to vapour.

Section 4.1 In this section in general you could do with better links (references to) your results section, so for instance refer back to the figures or sections where these results which you are bringing together are first mentioned. I also think this section could also be rather in the results section still.

L388: 'At all *of these* sites,'

L392: 'and QL' Usually increased QL (sublimation) would decrease melt? Or do you mean increased in terms of less negative? But I would expect the opposite if its windier.

Section 4.2: Again, to me this section is still results. You need to do the statistics here and show them  - are these relationships significant at a given p-value? What are the $R^2$ values? Just showing the scatters on their own in Fig 12 is not enough. Also consider looking at only the sites in the ablation zone or those in different regions separately.

L403: 'with latitude or altitude' - There does look to be a relationship with latitude, perhaps with an outlier? Do the stats to check.

L404: 'Neither average near-surface air temperature' - Again, Ta does look to relate to the cloud effect, but you need to do the stats to know!

Figure 12: Here it would really help (similar to in the line graphs above) to somehow differentiate (maybe using symbols) the different regions. You should also include a legend for this figure so the reader can understand it on its own. Also why not include also the influence of the turbulent fluxes and wind speed?

L435: Zongo's large seasonal variations in climate. Perhaps make it clear that the precipitation and cloudiness are the key variables which change seasonally here, rather than Ta.

L441-442: Is this reference to Chen et al. (2021) also referring to the site at BREW?

L458: 'in *the* first partial'

L470-473: You need to cite the studies you refer to here. Are you sure there are no studies that include changes in cloud in future glacier change?

L474-475: Reference here to your results figures/sections.

L476: 'during marginal melt seasons *and especially at high elevations*.'

L483 and 484 'metadata'

L488-489: 'As many…' You need to cite studies here, also I'm less sure what you mean, usually Swin is influenced strongly by topography whereas Lwin is less so (aside from cloud forming processes but they are not related to Swin). There are parameterisations of Lwin from Swin (e.g. Juszak and Pellicciotti, 2013), if that is what you mean?

L506: When mentioning the turbulent heat fluxes be clear about how the latent heat flux changes, since it is often negative.

L509-511: 'The association….' I think you could have pulled this apart in more detail, it feels like you have the data to understand this, but it needs more in-depth analysis than you have shown.

Data availability: Given your point in the limitations it would be much better if these data were made available together (with your analysis code) in a repository. Of course, it depends on the agreement of individual data providers, but you should aim for this.

Figure A1: Tidy up the labels here to use correct notation, also add units for the left hand variables.

Figure A2: Tidy up the labels and legend to use superscript (for units) and proper notation for the fluxes.

Figure A4/A6: Missing a legend, its not clear what the colours represent.

**References**

Juszak, I. and Pellicciotti, F. (2013) A comparison of parameterizations of incoming longwave radiation over melting glaciers: Model robustness and seasonal variability, *Journal of Geophysical Research: Atmospheres*, 118, 3066-3084, doi:10.1002/jgrd.50277

---

## Author Comment (AC1)

**Response to reviewer comment #1.** Note reviewers' text is shown in **blue**, with responses in **black**

Review of "Cloud forcing of surface energy balance from in-situ measurements in diverse mountain glacier environments" by Jonathan Conway et al. #tc-2022-24

General comments:

The authors presented a study on how cloud cover influences the glacier surface energy balance (SEB) in diverse climate settings across the globe based on observed meteorological data from 16 mountain glacier sites. They have compared the results and discussed the differences among 16 sites in terms of cloud's role in controlling local meteorology and SEB and thus glacier surface melt. First, the influence of cloud on the glacier mass balance is an extremely important topic to grow our understanding about physical glacier-atmosphere interactions where the existing community knowledge is poor. Second, this comparative study based on in-situ data from 16 diverse glacier sites is highly welcoming and has potential to increase our knowledge significantly. Finally, such understanding is also helpful to improve the SEB-based regional-wide glacier mass balance models and regional climate models. Based on my knowledge, the work is timely. Dataset selection/filtration steps are well-justified and standard, also methodology section is with strong physical background of the cloud cover estimation equations, etc. Figures are high quality and the statements and conclusions are well supported by the results/figures. Analysis of cloud effects on SEB across 16 sites is one of the unique part and gives a greater knowledge of spatial distribution of such understanding based on observed datasets. Also, I would like to commend the authors for this work which brought together several authors with similar interest and used the unique glacier SEB dataset in a common framework for better understanding the mountain glacier-atmosphere interactions. Also, they have nicely highlighted the future work needed and a need for a common repository of crucial AWS/SEB datasets.

I feel this is an interesting contribution to TC and valuable for the cryosphere community across the globe. The paper is interesting, concise and clearly written, therefore, I do not have many comments/suggestions for the authors as the manuscript is in already good shape. Below you will find some general and minor comments and suggestions that you might find helpful to improve the quality of the manuscript.

We thank the reviewer for their helpful and positive review. The suggested changes have helped clarify many points.

Specific comments:

1. L164: I understand the data scarcity of such high-quality glacier AWS datasets, but don't you think data of 10-days from a month is a bit under-represented the monthly features/statistics? Or how about half a month (15-days) considering limited data for each site, also could be better-justified? If you agree, I do not think it would take much time to consider.

Filtering using a lower limit of 15 days in a month would remove 3 additional site months, being June at CABL (12 days) and CACC (14 days), and November at MERA (14 days). At the Conrad Glacier sites (CABL and CACC) these extra records are in the melt season, so we prefer to retain them to increase the applicability of statistics calculated over the full melt season (e.g. Figures 6 to 13). For reference, only two site months were filtered by the 10-day criterion (being September at CABL (5 days) and CHHO (4 days)), with the other missing site months having no valid data.

2. In Figure 5, I am a bit surprised to see the overcast cloud fraction in July for CHHO/Chhota Shigri Glacier. It's very small, compared to other Himalayan glaciers (Yala, Mera etc.). However, I am aware that Chhota Shigri area is a monsoon-arid transition zone, where monsoon clouds don't penetrate much (Azam et al., 2014), but such small overcast cloud fraction days are surprising. Or is it due to not having complete-month data in July, as I see in Azam et al. 2014 it starts from 8 July 2013? In that case I believe it is quite under-represented. Otherwise, it is worth putting a small note on the figure caption about this.

The lower fraction of overcast periods at Chhota Shigri Glacier in Figure 5 is due to the use of daily average cloudiness and due the poor monsoon reach at Chhota Shigri Glacier compared to Mera and Yala glaciers, as pointed out by the reviewer. Mera and Yala glaciers (central Himalaya) are located in typical monsoon-dominated climate while Chhota Shigri Glacier (western Himalaya) is in Alpine climate (Azam et al., 2021, Science https://www.science.org/doi/10.1126/science.abf3668 ) and thus receives monsoonal precipitation occasionally. In our present analysis, we estimated the cloudiness at hourly scale and then estimated the mean daily values to reduce the noise, limit the influence of sub-daily variability and focus on synoptic scale (daily) variability in cloud-SEB analysis. On an hourly basis (Figure A4, there is a much higher fraction of overcast periods in July (25%) and August (40%).

We don't think the low cloud factor in July 2013 is because of short data (23 days). For instance, in Azam et al. (2014) the daily mean cloud factor was 0.34 for July 2013 (23 days) while it was 0.38 for August 2013 (complete month) and the mean of 60 days (selected period = 8 July to 5 Sept 2013) was 0.36.

We have amended the text to in section 3.1.2 to explain the reasons for fewer overcast periods at Chhota Shigri compared to other Himalayan Glaciers.

3. Section 3.4: How the melt pattern (frequency/amount etc.) gets influenced due to positive QL during summer in the Himalayan sites, for example, in Chhota Shigri, Yala, Mera, you get positive QL during summer, though it is not a significant amount. Is there any impact of clear-sky/overcast conditions in QL overall? Is it worth explaining briefly here?

Figure 11 shows the variation of QL with cloudiness confirming QL becomes positive or close to 0 during overcast periods at Chhota Shigri, Yala and Naulek. At Mera, QL is still negative, but less so than in clear-skies. Without a doubt this has an influence on the SEB and helps create increased QM in overcast conditions alongside increased LWin. We comment on this feature at lines 354 and 374.

4. Section 4.2: Although the authors choose to say that there is no relationship or not easy relationship between melt energy (QM) CE and latitude/air temperature, but from Figure 12, I think that the relationships are a bit clearer than with altitude or RH or SWin CE. In that case it is worth discussing briefly why there could be a bit clear relation between melt energy (QM) CE and latitude/air temperature? Or is it due to the higher latitude sites or maritime influence? Can you comment on this! This brings me to another important point: you should mention which glaciers are maritime/very high-altitude in Table 1, you can create a new column and mark them or find an easy

way. This will give a quick idea to the readers and they can correlate better among the diverse climate/sites.

We have added a column to table 1 to describe the regional grouping of glaciers. In Figure 12 we have added a legend to allow readers to identify the sites as well as linear correlation coefficients to each panel to aid in the interpretation. These show air temperature has a moderate and statistically significant association ($r=-0.54$, $p=0.03$) while latitude and altitude shows only weak and insignificant associations ($r=-0.25$, $p=0.32$ and $r=0.27$, $p=35$, respectively). We have also added further variables and discuss these relationships further in a new section 3.5 (see response to reviewer comment 2 for full text)

Figure and Table:

Figure 1: Here authors may cite the RGI 6.0 (RGI Consortium, 2017) in the caption, as they have used it in this figure, but I did not see any citation in the reference list. Also, you could mention the background image of this map (Natural Earth?).

Thanks. References have been added.

Table 1: Latitudes and Longitudes digits/decimals are not consistent, some with three decimals and some two. I would have made them two decimals for all sites.

The coordinates were supplied in a variety of formats (some very precise), so we have rounded to three decimals (~100 m) to make these more concise yet still precise enough to place the AWS within the general vicinity of the glacier. We would prefer to keep the precision where possible.

Figure 3: Does the colors mean anything? If yes, you can briefly mention in the caption, else you may remove the colors.

The colors align with the sections (2.2, 2.3, 2.4, 2.5) that are relevant. The figure has been updated with annotations and caption updated to "Steps used to process and analyse data, annotated with relevant sections of the methods."

Minor/technical comments:

L57: You could put Ts ≥ 0 °C or > -0.1 °C in parenthesis.

Changed to "When the surface is at the melting point (i.e. surface temperature ($T_s$) = 0 °C),"

L58: Please expand: w.e. (water equivalent) within the parenthesis, I see it first appears here in the manuscript.

Replaced with water equivalent

L70: What do you mean by highly reflective glacier surfaces? Do you mean fresh snow? You may mention a few as e.g.

Added 'clean snow' as an example

L71-73: Although the paper by Mandal et al. (preprint in TCD) is still in discussion stage, they showed that sublimation is also considerably reduced (about 2-3 times) due to clouds in the Chhota Shigri Glacier area-one of the sites in your present study.

Very interesting paper. We have added a reference to it in the introduction.

L99: I think you should expand the abbreviation of AWS here, because it first appears here and removes it from L107.

Done

L105: surface energy balance —> SEB

Done

L107-109: Remove 'balance' after radiation.

We have kept balance here to distinguish these components from other differentiations of radiation (e.g. diffuse, direct)

L124-125: NORD and CHHO are not expanded here, so it is a bit hard to follow, or can you cite Table 1 somewhere here, so that readers can quickly go to Table 1 and see what these short names refer.

Added to line 126 "(See Table 1 for site name abbreviations)."

L170: In equation 3, you have not mentioned about the sigma (σ)/ Stefan–Boltzmann constant. As you have mentioned details about all other variables/parameters, I would have mentioned for a bit easy read.

Added "where σ is the Stefan–Boltzmann constant ($5.67 \times 10^8$)."

L195: For partly cloudy, isn't it should be 0.2 > Nε < 0.8? Please correct.

Yes – corrected now $0.2 < N_\varepsilon < 0.8$

L204: Is it important to keep the last part of the sentence 'SWin does not provide meaningful values during the night time'? I would have deleted it because it is relatively understood that SWin is up only during daytime.

Good point. Have reworked sentence "In addition, cloudiness cannot be derived from *SWin* during the night time and terrain shading of *SWin* introduces further uncertainty, especially in winter."

L221: Can you cite someone here, as you say 'In studies of net radiation', where they defined CE as the difference between average and clear-sky conditions.

Citation added

L250: Partial cloud —> partly-cloudy, and elsewhere (e.g., L255, L265 and elsewhere).

Updated definition at line 197 to "partial-cloud as $0.2 < N_\varepsilon < 0.8$"

L244: In caption, add comma after εcs

added

L289: W m-2 —> W m-2

fixed

L289-291: Which three sites? It is difficult to identify them easily from 16 sites/legends, can you mention them in parenthesis?

Added to line 293 "(MIDT, MORT, CHHO)"

L303: Here you write melt-season, elsewhere it is melt season.

'melt-season' is used where the term is used as a compound adjective to define the period of measurement (e.g. melt-season air temperature), whereas 'melt season' is used when the term is used a noun (e.g. cloud conditions during the melt season. We have updated the text to ensure this is consistent.

We use wind climate to describe the typical conditions, rather than a specific weather system.

L341: I would have written clearly: In contrast to the percentage of hours with surface melt. Otherwise, for the general readers it is a bit rethinking what is fraction of time with melt.

Thanks, good suggestion. Changed to "In contrast to the percentage of hours with surface melt,"

L355: Evaporation because of Ts ≥ 0 °C or > -0.1 °C? You can quickly mention within the parenthesis for easy read for other/general readers.

Changed to "(indicating evaporation as $T_s$ = 0°C)"

L403-405: Here I think you should mention the colder and warmer sites, at least a few, in parenthesis. Else, until here the readers have already forgotten which sites are colder/warmer because there are 16 sites across the globe.

Examples of colder and warmer sites are now given in the text

L406-407: Is it proper to say it as correlation here? Or would you go with relationship, but again then relationship comes two times. You may find some other words. Cite Figure 12h after the albedo part.

Changed to "association"

L490: Conclusion does not have any section number. Maybe some formatting issue.

We have followed the journal template, which has no section number for the conclusion.

L491: How about, …. and surface energy balance over glaciers from very different…?

good suggestion. Changed to "surface energy balance over glaciers in very different climate settings"

Changed to "cloudy compared to clear-sky conditions."

L515: This sentence is similar to L480. Or you may remove it from L480 and combine it here in the Conclusion.

At the risk of being repetitive, it is worth restating this key result in both locations.

References:

Azam, M. F., Wagnon, P., Vincent, C., Ramanathan, AL., Favier, V., Mandal, A., and Pottakkal, J. G.: Processes governing the mass balance of Chhota Shigri Glacier (western Himalaya, India) assessed by

point-scale surface energy balance measurements, The Cryosphere, 8, 2195–2217, https://doi.org/10.5194/tc-8-2195-2014, 2014.

Mandal, A., Angchuk, T., Azam, M. F., Ramanathan, A., Wagnon, P., Soheb, M., and Singh, C.: 11-year record of wintertime snow surface energy balance and sublimation at 4863 m a.s.l. on Chhota Shigri Glacier moraine (western Himalaya, India), The Cryosphere Discuss. [preprint], https://doi.org/10.5194/tc-2021-386, in review, 2022.

RGI Consortium (2017). Randolph Glacier Inventory – A Dataset of Global Glacier Outlines: Version 6.0: Technical Report, Global Land Ice Measurements from Space, Colorado, USA. Digital Media. DOI: https://doi.org/10.7265/N5-RGI-60

Citation: https://doi.org/10.5194/tc-2022-24-RC1

---

## Author Comment (AC2)

**Response to reviewer comment #2.** Note reviewers' text is shown in blue, with responses in **black**

Review of Conway et al. Cloud forcing of surface energy balance from in-situ measurements in diverse mountain glacier environments

Conway et al. use a global selection of on-glacier AWS data to determine the effect of clouds on the surface energy balance. They investigate the influence of clouds on the near-surface meteorology, individual energy fluxes and the frequency and magnitude of melt. They found an increase in the frequency of melt during cloudy conditions but the effect of clouds on the energy available for melt varied spatially.

Overall, I think the purpose of the paper is a very good one, and it is certainly an interesting approach to look at the impacts of clouds across a range of sites, since often energy balance studies are confined to one site or a region, so the global aspect is appealing. The paper is also clearly written throughout, and the methodology followed is sensible. However, I do have a couple more significant concerns which I highlight in the major comments below:

We thank the reviewer for their thorough and insightful review, which has helped us improve the manuscript. As well as addressing the minor comments, we have added two sections to the manuscript that address the main concerns listed below.

Major comments

1. Depth of analysis and understanding of global trends

In the main results section the variation in the results (in terms of the cloud effect on the meteorology, surface fluxes and melt) caused by the different location, climate and elevation of the sites is mentioned (certainly when the effect is quite clear). However, I don't think the authors really make the best use of their dataset to fully interrogate the spatial variation in the results. The relationship between the cloud effect and station and energy balance characteristics is not investigated fully (with scatter graphs) until section 4.2 (which anyway should be a result section). Although the figures earlier in the results section are clear as they are they are not well suited to investigating the spatial differences and improving these figures to make the station characteristics clearer and including scatter graphs earlier in the results would be a good idea. Furthermore, the analysis in section 4.2 is not robust, the authors need to calculate the correlation and regression (if appropriate) coefficients and report them in the paper. Currently assessments are made only on a visual assessment. In general, I think the paper needs an extra stage of analysis yet to give its findings more credibility, and this may also allow clearer findings on why the effect of clouds on melt energy varies in sign spatially.

We have moved the previous section 4.2 into the results section (now section 3.5) and expanded the analysis to include an objective assessment of correlation and statistical significance, as well as analysing further variables for their relationship to cloud effects on melt energy in Figure 12. We have also added a legend to Figure 12 that allows readers to identify the station characteristics of each site, as well as regional climate groupings to Table 1. Together this presents a more thorough assessment of the spatial variations of cloud effects.

New section 3.5:

""“

**3.5 Relationships between $Q_M$ cloud effect and site characteristics**

While the average change in $Q_M$ with cloudiness is small at some sites, it is instructive to assess whether the melt-season average $Q_M$ cloud effect (CE) at the various sites can be related to geographic or climatic parameters. Figure 12a,b shows the average relationship between cloudiness and melt at the various sites does not follow easy relationships with latitude or altitude. Average near-surface air temperature is moderately correlated to $Q_M$ CE (Figure 12c). Sites with lower $T_a$ (e.g. MERA, KERS) generally have smaller $Q_M$ CE than sites with higher high $T_a$ (NORD, CABL, MORT), but with some notable exceptions (e.g. LANG has positive $Q_M$ CE with relatively high $T_a$). Average cloudiness shows some association to $Q_M$ CE with clearer sites tending to have more negative $Q_M$ CE (Figure 12e), with the exception of Tropical sites with predominately clear-skies (KERS, GUAN) that show neutral $Q_M$ CE. Neither, average wind speed or relative humidity show a clear relationships with the $Q_M$ CE (Figure 12d,f). Average turbulent heat fluxes and *LWin* are moderately correlated $Q_M$ CE (Figure 12g,h,j), largely following the pattern of sites shown for $T_a$, while average *SWin* is not significantly correlated (Figure 12i).

Considering the association of radiative and melt cloud effects, average incoming radiation cloud effects explain some of the variance of $Q_M$ CE, with *LWin* (Figure 12l) showing a stronger association than *SWin* CE (Figure 12k). Combined, the incoming radiation cloud effects can explain over half (53%) of the variation in $Q_M$ CE (Figure 12m). Surface albedo has a similar correlation to $Q_M$ CE (Figure 12n) as the incoming radiation cloud effects together. The combination of these into the *Rnet* CE shows the clearest relationship to $Q_M$ CE (Figure 12o). In general, sites that experience a radiation paradox (LANG, ZONG, MERA) also experience greater melt in cloudy conditions (positive $Q_M$ CE), while sites with negative *Rnet* CE experience less melt in cloudy conditions (Figure 12o).

Turbulent flux cloud effects are also moderately correlated to melt-season average $Q_M$ CE (Figure 12p,q) and when combined explain a 44% of the variance in $Q_M$ CE (Figure 12r). Thus, sites where $Q_S$ decreases with cloudiness show more negative $Q_M$ CE. Sites where $Q_S$ varies little with cloudiness and/or $Q_L$ becomes less negative/more positive during cloudy periods show neutral or positive $Q_M$ CE. Interestingly, the radiative and turbulent heat cloud effects show a moderate association, with sites with large negative *Rnet* CE also having a negative net turbulent flux cloud effect, and vice versa (Figure 12s).

[Figure]

**Figure 12: The variation of average melt-season $Q_M$ cloud effect (CE) with (a) station altitude, (b) absolute station latitude, average melt-season (c) $T_a$, (d) $RH$, (e) $N_\varepsilon$, (f) wind speed ($WS$), (g) $Q_S$, (h) $Q_L$, (i) $SWin$, (j) $LWin$, (k) $SWin$ CE, (l) $LWin$ CE, (m) $SWin+LWin$ CE, (n) albedo, (o) $Rnet$ CE, (p) $Q_S$ CE, (q) $Q_L$ CE, (r) $Q_s + Q_L$ CE. (s) is variation of $Rnet$ CE with $Q_{S+}Q_L$ CE. See Section 2.5 for definition of CE. Average melt-season values are calculated by averaging values from the 5 cloudiness bins equally. The Pearson correlation coefficient (r) and associated p-value are given above each panel.**

*"""*

The aim of the present paper is to describe the relationships between cloud, meteorology, SEB and melt at different sites and discuss what features are common or unique across the sites. A deeper assessment of why these relationships differ between sites requires further analysis that is beyond the scope of the present paper i.e. it would require an investigation of how regional climatology impacts cloud characteristics and the co-variance between cloud radiative forcing and changes in near-surface meteorology during cloudy conditions at each site – analysis that would be better suited to a separate paper.

**2. Discussion**

The first two sections of the discussion are really still results, and then there are only the limitations and implications sections, with the latter section really missing references. I miss here a proper in-depth discussion which brings together the understanding alongside other studies. Consider

answering some questions, e.g. Why do clouds influence the near surface meteorology in the ways you found? (explain the physical mechanisms) What factors cause changes in the impact of clouds on melt? You touch on this in sections 4.1 and 4.2 and also in your limitations, but I feel you need to set this out more as a clear discussion with each of the factors and their influence. There is also a need to bring in other energy balance studies which have looked at the cloud effect (or even seasonal differences where cloud cover likely varies markedly), especially in regions not covered by your analysis.

We have added a section to the discussion "4.3 Mechanisms influencing SEB changes with cloud" in which we discuss the physical processes leading to changes in SEB due to cloud as well as linking this work to other studies:

""

**4.3  Mechanisms influencing SEB changes with cloud**

In addition to the key role that surface albedo plays in determining $Rnet$, there are three key mechanisms that drive temporal changes in SEB with cloudiness

i)      direct forcing of incoming radiation (decreased $SWin$ and increased $LWin$),

ii)     changes to near-surface meteorology that alter turbulent heat fluxes

iii)    surface and subsurface temperature feedbacks that alter net radiative and turbulent fluxes

Here we demonstrate that direct forcing of incoming radiation and surface albedo explains much of the net effect of clouds on $Q_M$ across sites. The high correlation between melt-season average $Rnet$ CE and $Q_M$ CE between sites (Figure 12), along with the sensitivity of these averages to the length of the melt season (Figure 10 vs Figure 12) underlines the primary control of direct and indirect radiative mechanisms on determining the sign of melt response to cloud. It is likely that substantial seasonal variations of $Rnet$ CE exert the primary control on the effect of clouds on glacier melt.

Changes in turbulent heat flues with cloudiness tend to be smaller in magnitude than changes in $Rnet$ (Figure A5), except for the more extreme cases where air temperature changes greatly with cloudiness, e.g. NORD, where $Q_S$ markedly decreases with cloud and MERA, where $Q_L$ becomes far less negative during cloud. Despite this, net turbulent heat flux cloud effects show moderate correlation to $Q_M$ CE, and thus changes in near-surface meteorology play a significant role in determining the net response of melt to cloud. These findings echo those of Liu et al. (2021) who show increased melting during cloudy periods on Mt Everest are due to increased $Rnet$ as well as lower wind speeds that drive smaller losses to $Q_L$, and Conway et al. (2016) who found changes to $Q_L$ contributed to increased melt during cloudy periods. Future work should assess the mechanisms driving the observed covariance between cloudiness and near-surface meteorology at different sites, e.g. Do large-scale changes in airmass or local/meso-scale processes drive changes in $T_a$ with cloud? How well are these processes represented in the datasets used to force glacier melt models on regional scales? Seasonal changes in the relative magnitudes of turbulent and radiative cloud effects also deserve further scrutiny.

Surface temperature responds quickly to changes in SEB, and here we show that during cloudy periods, a melting state is observed more frequently, in line with previous research on maritime glaciers (Conway et al., 2016). We have not attempted to analyse further surface and sub-surface temperature feedbacks here as not all datasets contain these variables and a detailed analysis is

more suited to sensitivity experiments that allow the transient response of sub-surface temperature, humidity and refreezing to the resolved.

The increased frequency of melt during cloudy conditions, especially at higher elevations, raises the question of how glacier-wide melt is altered by clouds, along with how glacier-wide surface mass balance is altered by refreezing. Van Tricht et al., (2016) show increased runoff from the Greenland Ice Sheet during cloudy periods due to increased melt extent and decreased refreezing of melt water, while Niwano et al. (2019) found clouds increase melt extent but reduce total melt due to feedbacks between cloudiness and near-surface humidity. These studies are in line with the findings here – that clouds enhance the possibility of melt at a given site, by removing large negative *LWnet* and $Q_L$ fluxes to precondition the surface to melt, but do not necessarily cause greater melt unless albedo is high enough to cause a radiation paradox or unless increased near-surface air temperature, humidity and/or wind speed causes an increase in net turbulent fluxes.

„”"

Minor comments

L56: To me the subsurface is more the material under the glacier, I think Qc is the conductive heat flux into the surface.

Changed to "$Q_C$ is the heat flux at the surface from conduction within the glacier"

L57: Are all the fluxes in W m-2 (they usually are but just state this)?

Yes, and now noted in text

L86-88: It might be worth expanding on some of these methods to derive cloudiness and their main assumptions/difficulties.

We expand on the methods to derive cloudiness in Section 2.3.

L99: Define AWS here.

done

In methods: Maybe it would be useful to have a simple idea of the climate type/seasonality of each site? It seems that the overall broader climatology will drive the differences in the cloudiness patterns and how they relate to the melt seasons, so having this context early on would be useful.

We have introduced the broad climatology at the end of Section 2.1

"Figures A1 and A2 show monthly average meteorology and SEB fluxes for each site used in the analysis. A few broad groupings of sites (listed in Table 1) can be identified through seasonal trends in near-surface air-temperature ($T_a$; °C) or relative humidity (*RH*) in Figure A1: mid- and high- latitude maritime and continental sites with strong seasonal cycles of $T_a$ but small variations in *RH*; Himalayan sites with strong cycles of $T_a$, and distinct wet and dry seasons; Tropical sites with small variations in $T_a$ and distinct wet and dry seasons; and a mid-latitude arid site (GUAN) with low *RH*."

L110: Due to the different methods of the calculation of the turbulent fluxes consider including a table with how the non-radiative fluxes were calculated for each site (in the appendix/SI would be fine).

We have clarified that the turbulent fluxes were all calculated using bulk aerodynamic methods. We list the papers corresponding to each dataset, so the reader can find extra details if they require.

Figure 1: Maybe include insets where you have several sites relatively close by in a region. Also use a different symbol to label colour for readability.

Thanks for the suggestion, but we prefer to keep the figure as is.

Sections 2.3, 2.4 and 2.5 – if these are all within 'data processing' then it might make sense for these sections to be 2.2.1, 2.2.2, 2.2.3 (so sub-sections of 2.2)

Thanks for the suggestion but we prefer to keep the original sections

Figure 3 caption: 'Steps'

done

L149-162: Honestly, I think this could go into an appendix or SI. But I am wondering, if you had to do these quality checks then how do you know that the SEB fluxes were also calculated post these quality checks - I thought you were using the published data (which hopefully would already be checked?) Can you clarify this please. Calculating Ts from LWout works quite well but not always, give the reference for how you did this.

The quality control step was mainly to ensure the datasets were homogenous in terms of temporal coverage and the variables included. Because the full time series data from each site was provided, and in some cases this included periods that different variables were not available, these periods had to be removed. We have given a clarified we use the Stefan-Boltzmann law and *LWout* to calculate surface temperature if it is not available.

L172: Also define sigma as the Stefan Boltzmann constant, and give a reference for this equation.

Done

Section 2.4 could probably be shortened.

For the sake of completeness we have retained the text.

Figure A2, A4: Please add a legend so the reader knows that the colours are used to define the melt season.

Consistent with Figure 5, a note has been added to the caption of Figure A4 and A6: "Months defined as within the 'melt season' are shaded blue."

L224: Why only look at the cloud effects versus the radiative fluxes? You do look in Figure 11 at the importance the turbulent fluxes for melt and how that varies with cloudiness, but why not also include them as for the radiative fluxes in Figure 7. Furthermore, is there not some circularity in looking at the LWnet differences given that LWin is used to calculate the cloudiness?

We have followed previous research that has analysed the direct effects (radiative) of clouds separate from the indirect effects (through temperature/humidity/wind etc). In Figure 11/A5 we analysis the variation of the turbulent fluxes with cloudiness.

We agree there is some unavoidable degree of circularity in analysing radiative fluxes that have been used to derive cloudiness. However, as LWin does not solely depend on cloudiness, but also the variation of temperature and humidity, the circularity is not complete. For instance, at Brewster Glacier, the increase in LWin between clear-sky and overcast conditions is approximately the same as the change in clear-sky LWin due to seasonal variations in air temperature. The method used to

calculate cloudiness removes the effect of temperature/humidity on LWin, so the effect of these variations in near-surface meteorology on LWin is retained in the analyses in Figures 6 and 7.

Figure 4 caption: It would be useful to have a legend, or at least to explain what the darker via lighter colours represent. From the text it seems like darker colours = greater frequency of conditions, but this should also be clear from the figure/legend on its own. Consider outer boxes (or other methods) showing the splits between regions.

The caption has been updated to make it clear the plot shows the frequency of different conditions along with the meaning of different colors.

L258: 'between monsoon and arid regions although it still shows an increase in partially cloudy conditions in the melt season' Or something similar, just for clarity.

Agreed – changed to "between monsoon and arid regions, though the fraction of partial-cloud conditions still increases in July and August."

Figure 5: This is nice way to show the cloudiness at the sites, but it might be useful to have some overall metrics so its slightly quicker to compare sites, e.g. the mean and range of melt season monthly cloudiness? It might also be useful to group by region (Himalayan, European etc.) Even though I know most of these sites and where they are its not so easy to see trends, and I imagine it would be harder if you didn't know inherently the site locations.

The sites are ordered by latitude in this and other figures, so naturally fall into regional groupings. We now show average melt-season cloudiness against cloud effects in Figure 12.

Figure 6: Maybe it would be helpful to take this a step further, for instance can you relate the gradients of these lines to the site lat/long/elevation, e.g. in a scatter graph? It's easy to see that KERS, MERA and ZONG are different but harder to know what is causing the variation in the other sites. You do attempt this in the discussion but I think this analysis could be more thorough and come earlier in the paper.

We did investigate analysis of gradients of relationships as suggested but settled on the overall cloud effects. An analysis of gradients could be undertaken in future work.

L289: 'cloud effect is small and negative' It would be useful also to scatter this overall change in Rnet against the sites to see if there are regional clusters.

This is done in Figure 12.

L293: 'more positive response to' - do you mean in terms of an increase in Rnet here?

'more positive and/or less negative *Rnet* cloud effect'

L307: 'relationship is weak and non-linear' - Quantifying the strengths of these relationships and their gradients (for all the variables in Figure 8) would be a good idea.

We have clarified the patterns displayed at these sites in Figure 8 "…, QASI, which shows no large change cloudiness and CACC, which shows peak wind speed at moderate cloudiness."

L323: 'at all study sites' - this doesn't appear to be the case for GUAN.

Have added "with the exception of GUAN, which experiences very infrequent melt in all conditions."

L329 – 331 'While.....day and night' – Add a reference to this effect if you don't show the analysis yourself.

We have modified the sentence to read, "the higher percentage of hours with melt during overcast conditions indicates that night time melt is more frequent during overcast periods."

L331- 334: This sentence might be better earlier in the paragraph.

Thanks, but we prefer to keep the original position

Figure 9: Here and in other similar figures, it might be a good idea to use different line styles as well as colours to indicate different regions? It might also allow you to use fewer colours, and have a palette which is more colour blind friendly. The strength of this paper is the wide range of sites and yet you need to show better the regional/climate/elevation differences in your plots.

Thanks for the suggestion, but we prefer to retain consistent line style and not pre-determine the expected groupings in the results too much. As the sites are grouped by latitude, the groupings of colors do follow this.

L337: 'indicating sublimation' Since we are going from ice to vapour.

Because this analysis in Figure 11 is limited to timesteps where the surface is melting, QL results in evaporation of melt water. We have clarified in the text "(indicating evaporation as $T_s$ = 0°C)"

Section 4.1 In this section in general you could do with better links (references to) your results section, so for instance refer back to the figures or sections where these results which you are bringing together are first mentioned. I also think this section could also be rather in the results section still.

Thanks. We have added references to the appropriate figures but have kept this section in the discussion as it does not introduce new results but rather attempts to synthesize the results.

L388: 'At all of these sites,'

corrected

L392: 'and QL' Usually increased QL (sublimation) would decrease melt? Or do you mean increased in terms of less negative? But I would expect the opposite if its windier.

Here we define QL as a positive flux towards the surface, so increased means less negative or more positive. The text has been clarified.

Section 4.2: Again, to me this section is still results. You need to do the statistics here and show them - are these relationships significant at a given p-value? What are the R2 values? Just showing the scatters on their own in Fig 12 is not enough. Also consider looking at only the sites in the ablation zone or those in different regions separately.

Correlation statistics (r and p values) have now been added to the figure, to objectively assess which relationships are statistically significant.

L403: 'with latitude or altitude' - There does look to be a relationship with latitude, perhaps with an outlier? Do the stats to check.

QM cloud effect has a linear correlation of $r$ = -0.25 to latitude and $r$ = 0.27 to altitude, so neither variable explains much. Figure updated with correlations.

L404: 'Neither average near-surface air temperature' - Again, Ta does look to relate to the cloud effect, but you need to do the stats to know!

Yes, air temperature is indeed moderately correlated with QM cloud effect ($r$ = -0.54). Figure and text updated.

Figure 12: Here it would really help (similar to in the line graphs above) to somehow differentiate (maybe using symbols) the different regions. You should also include a legend for this figure so the reader can understand it on its own. Also why not include also the influence of the turbulent fluxes and wind speed?

Figure has been updated with legend, as well as wind speed and the turbulent fluxes.

L435: Zongo's large seasonal variations in climate. Perhaps make it clear that the precipitation and cloudiness are the key variables which change seasonally here, rather than Ta.

Modified to "large seasonal variations in precipitation and cloudiness"

L441-442: Is this reference to Chen et al. (2021) also referring to the site at BREW?

No, this is for a Laohugou Glacier No. 12 in China – now clarified in the text.

L458: 'in the first partial'

corrected

L470-473: You need to cite the studies you refer to here. Are you sure there are no studies that include changes in cloud in future glacier change?

References added. To our knowledge, the only global study to include changes in radiation is Shannon et al., 2019.

Shannon, S., Smith, R., Wiltshire, A., Payne, T., Huss, M., Betts, R., Caesar, J., Koutroulis, A., Jones, D., and Harrison, S.: Global glacier volume projections under high-end climate change scenarios, The Cryosphere, 13, 325-350, 10.5194/tc-13-325-2019, 2019.

L474-475: Reference here to your results figures/sections.

Sentence removed as largely repeated sentences on either side.

L476: 'during marginal melt seasons and especially at high elevations.'

Thanks, and modified

L483 and 484 'metadata'

Corrected

L488-489: 'As many...' You need to cite studies here, also I'm less sure what you mean, usually Swin is influenced strongly by topography whereas Lwin is less so (aside from cloud forming processes but they are not related to Swin). There are parameterisations of Lwin from Swin (e.g. Juszak and Pelicciotti, 2013), if that is what you mean?

Yes, these are the sorts of parameterisations referred to. Glacier mass balance models use these to compute LW/SW at specific elevations, or to calculate LW/SW from near-surface meteorology (i.e. air temperature, humidity). Now clarified in the text:

"As many glacier SEB models rely on empirical relationships between *SWin* and *LWin* to modify these variables to account for local-scale changes in near-surface meteorology (e.g. Mölg et al., 2009a; Conway et al., 2015),"

L506: When mentioning the turbulent heat fluxes be clear about how the latent heat flux changes, since it is often negative.

Have added 'Less negative and/or more positive' in front of 'turbulent latent heat fluxes'

L509-511: 'The association….' I think you could have pulled this apart in more detail, it feels like you have the data to understand this, but it needs more in-depth analysis than you have shown.

We have clarified the associations of clouds and melt energy in an expanded results section 3.5 and discussion section 4.3. We agree there is scope for further analysis, but it is beyond the scope of this manuscript.

Data availability: Given your point in the limitations it would be much better if these data were made available together (with your analysis code) in a repository. Of course, it depends on the agreement of individual data providers, but you should aim for this.

We prefer to work towards a common repository in the future, rather than publish the collated dataset now. In the interim, the collated dataset could be made available for analysis to individuals on request (and agreement of the providers). The code to read in and analyse individual datasets will be made available online.

Figure A1: Tidy up the labels here to use correct notation, also add units for the left hand variables.

done

Figure A2: Tidy up the labels and legend to use superscript (for units) and proper notation for the fluxes.

done

Figure A4/A6: Missing a legend, its not clear what the colours represent.

Figure captions have been updated to clarify the meaning of colours and shading.

References

Juszak, I. and Pellicciotti, F. (2013) A comparison of parameterizations of incoming longwave radiation over melting glaciers: Model robustness and seasonal variability, Journal of Geophysical Research: Atmospheres, 118, 3066-3084, doi:10.1002/jgrd.50277